# Archean rifts and triple-junctions revealed by gravity modeling of the southern Superior Craton

C. Galley [1,2] ✉, M. Hannington[1], E. Bethell [1], A. Baxter [1] & P. Lelièvre[3]

The nature of Archean tectonics and the associated geodynamic regimes are much debated in modern geoscience, despite decades of research. In this study, we present a geophysical model to show that, by the Neoarchean, convective forces from rising mantle plumes or early forms of plate subduction caused widespread extension, creating linear zones of crustal growth. These regimes can be identified as Archean rifts in the ancient rock record by the topography of the Moho, i.e., a shallowing of the boundary between the crust and the lithospheric mantle. Gravity data collected over the Abitibi greenstone belt, a particularly well-preserved portion of Neoarchean crust located in Canada's Superior Province, was modeled to produce a topographic map of the Moho. The model shows corridors of shallow Moho surrounding islands of thick, intrusion-filled crust and is interpreted to be a snap-shot of microplate growth and breakup between 2.75 to 2.69 Ga. The connectivity of the interpreted relict rifts is possible evidence for the existence of Neoarchean plate boundaries and triple junctions and supports a model of at least local mobile-lid tectonics during this stage of Earth's history.

During the Archean the Earth's mantle was hotter than it is today, leading to thicker crust and likely different tectonic processes from those observed now[1–3]. A proposed lack of mobile-lid tectonics was suggested, in part, from an apparent lack of Archean rifting[4]. Mantle plumes and convection are thought to have driven much of the crustal extension in the Meso- to early Neoarchean, spanning 3.2–2.5 Ga[5–7], with mafic material infilling early microcontinental fragments[8–11]. However, extension and crustal infilling does not necessarily constitute crustal rifting. The Meso- to Neoarchean is seen by some as a pivotal time in Earth's history, that was potentially transitional between stagnant-lid and mobile-lid tectonics[2,12]. The timing of the onset of global mobile-lid tectonics is a contentious issue, with some authors suggesting it was a gradual and spatially heterogeneous process across the Earth's surface[2,12]. This study investigates the Abitibi greenstone belt, a Neoarchean terrane where mobile-lid tectonics may have been active but the architecture of the ancient rifts is not well established[13].

In mobile-lid tectonics the crust comprises of many rigid or semi-rigid lithospheric plates[12,14], bounded by divergent, convergent, and/or transform margins. The plates are widely affected by horizontal displacements and by break-up and growth through rifting[12,14]. Driving mechanisms for the formation, breakup, and displacement of the plates or microplates are thought to include subduction[12] or plume-induced mantle convection[6,7,15]. Bird (2003) defined modern microplates as small, mostly rigid areas of lithosphere that range in size from 1000 s up to 1,000,000 km[2][16]. Detailed studies of today's oceanic microplates show that they behave as independent plates but are continuously deforming structures[17,18].

The Abitibi greenstone belt, situated in the southern Superior Province of Canada, is the largest and one of the best preserved Neoarchean greenstone belts in the world[19] (Fig. 1). It formed between 2.75 and 2.69 Ga in a series of submarine volcanic events thought to have been centered in Archean rift basins. Most authors agree that the east-west trending, predominantly mafic volcanic units were emplaced

[1]University of Ottawa, Ottawa, ON, Canada. [2]Memorial University of Newfoundland, St. John's, NL, Canada. [3]Mount Allison University, Sackville, NB, Canada.
✉e-mail: gchrist2@uottawa.ca

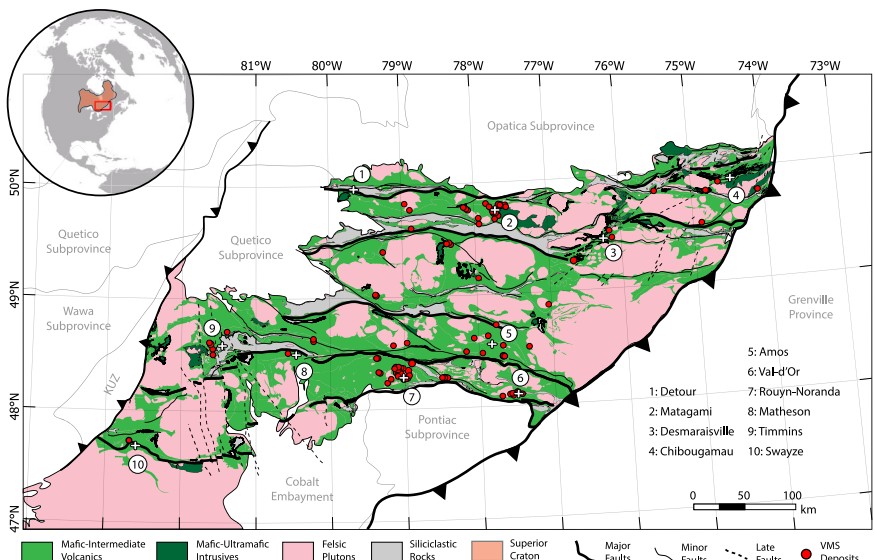

**Fig. 1 | A geologic map of the Abitibi greenstone belt[70], hosted within the Superior Craton of North America. Towns/mining districts of note are numbered.** The map is displayed using the NAD 1984 UTM 17N projection. KUZ— Kapuskasing Uplift Zone; VMS—volcanogenic massive sulfide deposits. Major faults are regional-scale transcrustal features (i.e., shear zones) reactivated by late-stage north-south shortening[22].

along similarly oriented rifts that formed by north-south extension[20–23]. However, the geometry of the rift network and the number of rifts are uncertain. In particular, the present-day surface expression of the rifts is obscured by folding, thrusting, uplift, and erosion. A deeper image is required to identify the rift architecture below the upper crustal deformation. In this study, we show signs of crustal thinning and rifting preserved as extensional scars at the interface between the crust and lithospheric mantle, i.e., the Moho[24,25]. Crustal extension and thinning brought dense mantle closer to the Archean Earth's surface. As a result, the zones of crustal thinning retain an anomalously high gravity signature that can be modeled as the ancient topography of the Moho[24–27]. Although rifting by itself does not prove a large-scale mobile-lid tectonic regime, the existence of crustal fragments that are encircled by rifts are a strong indication of at least a local microplate regime in the Abitibi.

The original lithostratigraphic assemblages of the Abitibi greenstone belt have remained largely intact since the end of the Archean[19]. The two main stages of tectonic deformation since 2.69 Ga were: 1) N-S compression followed by 2) E-W shearing[28]. The N-S shortening thickened the crust by upper crustal imbrication and rotated the near-surface volcanic units that lie above the brittle-ductile transition zone. Seismic profiles have shown that this imbrication does not extend to the Moho, thus preserving the Archean Moho topography in relation to the upper crustal units[29]. Many features of the upper crustal rocks are widely distributed by syn-tectonic intrusions, but certain elements observed at surface correlate spatially with the inferred rifts. These include the spatial distribution of volcanogenic massive sulfide (VMS) deposits[30] and geochemical whole-rock and time-resolved isotope data[20,21].

In this work we present a gravity-derived model of the Moho surface beneath the Abitibi greenstone belt that reveals a possible Neoarchean rift network. The proposed geometry of the microplate mosaic, supported by trace element and isotopic data of associated volcanic rocks and the presence of rift-related mineral deposits, shows an empirical relationship between mantle plume activity, microplate formation, and the development of major mineral systems within a local mobile-lid tectonic environment.

## Results
### Gravity modeling of the Moho
To model the Moho topography in the Abitibi, we inverted Bouguer gravity data compiled by Natural Resources Canada[31] (Fig. 2; see "Methods" for details on data processing) to produce a three-dimensional (3D) density model of the Earth's crust (Fig. 3). The inversion modeling approach we took created a petrophysical model by iteratively solving for a density distribution in the subsurface that created a gravity signal closely resembling the measured gravity data. The Moho interface was then extracted from the 3D density inversion model by isolating the 3070 kg/m³ iso-density surface. Extracting an iso-density layer from a 3D density model has been shown to be an accurate proxy for the Moho when performing minimum-structure inversion modeling[25,26,32]. Previous studies similarly modeling the Moho have used the 3020 kg/m³ iso-density contour[25,26,32], however through further testing we found that the 3070 kg/m³ contour better fit known Moho depths in the Abitibi. In contrast to the minimum-structure style inversion modeling used in this study, a more common approach is to use the surface-based Parker-Oldenburg modeling method to model the Moho[33,34]. This surface-based method solves for the depth of an interface between an assumed homogeneous-density crust (e.g., 2850 kg/m³ [27]) and the mantle (e.g., 3330 kg/m³ [27]). Homogeneity in the crust and mantle is generally assumed after applying wavelength filtering to the gravity data aimed at removing the signal resulting from density variations in the crust and upper mantle, isolating the signal from the Moho[35]. We opted not to use the surface-based modeling method, instead using the more generalized, albeit more computationally demanding, minimum-structure inversion method (see "Methods" for details), to account for the known heterogeneity in the density of the Abitibi crust[36]. However, a noted strength of surface-based modeling methods is that they solve for discrete interfaces. The Moho under the Abitibi, as well as in other Archean terranes, is believed to be a sharp boundary[37] and should ideally be modeled as such. Therefore, we implemented a two-step modeling procedure that first solved for the 3D density distribution, as described above, from which a representation of the Moho was extracted in the form of an iso-density surface. The next step was to rerun the inversion fixing the modeled surface of the Moho and

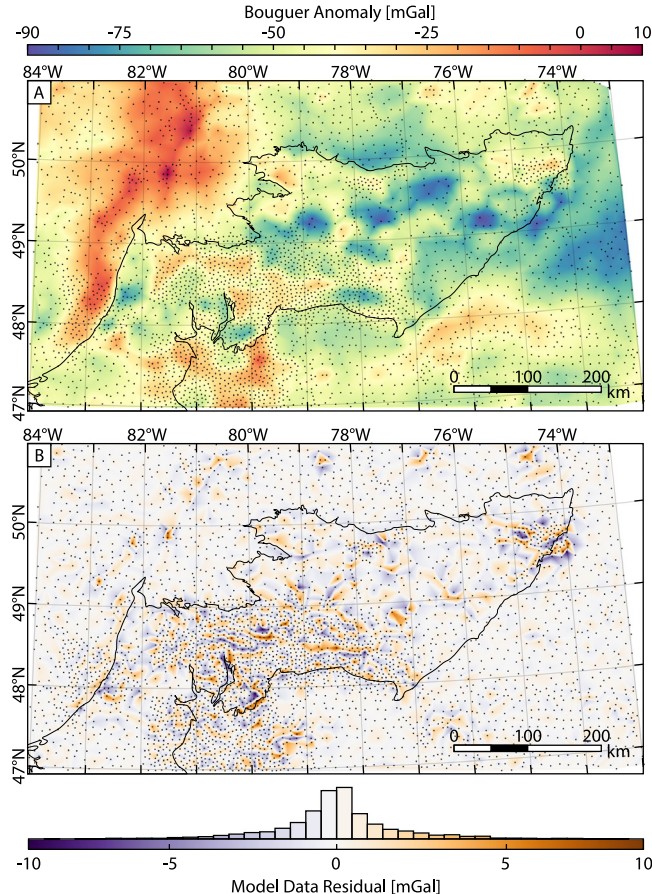

**Fig. 2 | Bouguer gravity anomaly map of the Abitibi, derived from data compiled by Natural Resources Canada[31].** **A** The Bouguer data map and **B** the difference (residual) between the forward signal produced by our inversion model and measured Bouguer data. Black dots in both figures represent the measurement locations for the gravity data, with the Abitibi outlined in black[70].

separately constraining the densities in the crust and upper mantle as to better agree with petrophysical measurements from the Abitibi[38] (see "Methods" for more details). The final Abitibi 3D density model then included a discrete density change across the Moho. The smooth-nature of the initial 3D inversion model allowed for an uncertainty in Moho depth (approximately ±3.5 km) to be quantified using a ±20 kg/m³ uncertainty in model density[25].

A reference model of 0.0106*z kg/m³ was used to both resolve the Moho and a model with increasing density with depth[39]. The slope of this reference model was determined through iterative testing (see "Methods").

### Comparison with seismic profiles

The results of the gravity inversion are compared to Moho depths from the Lithoprobe reflection seismic experiments[40], a Lithoprobe seismic refraction model[41,42], those from the CRUST 1.0 model[43], and a teleseismic model[44] (Fig. 3B, C). The reflection seismic sections were depth-migrated using a constant velocity model of 6 km/s, creating an approximate depth model of the Moho and overlying crustal features[40]. The Lithoprobe seismic refraction line modeled by Winardhi and Mereu[42] resolved a Moho line from a 2D velocity model of the crust and upper mantle, providing more accurate depth information on the Moho. The teleseismic measurements were inverted to solve for the velocity structure of the crust, and their radial receiver functions were plotted to reveal the Moho[44]. The CRUST 1.0 model is the coarsest of the Moho models with a spatial resolution of 1 arc-degree (~111 km) and Moho depths derived from a velocity model of

generalized Archean crust (upper crust: 6.2 km/s, middle crust: 6.4 km/s, and lower crust: 6.8 km/s)[43]. As such, the CRUST 1.0 model is used as a comparison for the general depth to Moho within the Abitibi, as the model is too coarse to resolve any short-wavelength features in the Moho.

A close fit is observed between our model and previously published Moho depths (Fig. 3). The closest fit is seen between the Moho derived from the teleseismic model and the refraction seismic model's Moho (Fig. 3C). While resembling the Moho in our gravity model, the Moho derived from the reflection seismic section provided a worse depth fit of the seismic models when used as a comparison (Fig. 3B). In particular, there is a discrepancy between the models within Line 12, where the gravity model shows a shallowing of the Moho, but the reflection model shows a deepening. This offset likely results from the depth information of the reflection profile being calculated by assuming a constant velocity throughout the crust. As this deepening of the Moho is only seen in Line 12 of the reflection profile, and does not exist in the gravity, teleseismic, or refraction models, we conclude that it is likely a distortion due to the assumed homogeneity of crustal velocities.

Whereas earlier models of the Abitibi Moho have been mostly two-dimensional (i.e., the Lithoprobe and teleseismic sections) or low-resolution (i.e., the CRUST1.0 model; 1 arc-min spacing), our model provides both 3D and higher resolution (10 km) information about the Moho topography.

### Crustal density structure

Our 3D density model of the Abitibi greenstone belt shows a range of densities, correlating with known volcanic and intrusive domains (Fig. 3). The exposed felsic plutons have a lower density than the surrounding mafic to intermediate volcanic rocks, as they contain more $SiO_2$ (i.e., quartz and feldspar) and less of the denser MgO- and FeO-bearing minerals (i.e., amphiboles, biotite, and Fe-Ti oxides). The calculated depth of the Moho is shown in cross-sections through the density model (Fig. 3B, C). One of the strengths of 3D density modeling is the ability to separate near-surface features in the model (Fig. 3A) from those at depth (e.g., undulations in the Moho; Fig. 4A)[45]. As such, the visual correlation between the presence of high density mafic volcanics at the surface of the 3D model to the location of the modeled rifts is likely a result of a geologic process whereby these volcanics were created at these rifts and not the result of a modeling artifact. The results show that the density of the mantle below the Abitibi, up to 70 km depth in our model, is strongly layered and increases sharply with depth. Above the Moho, the crust continues to be strongly layered for approximately 10 km, which we interpret as the high-density, seismically reflective lower crust[40]. The mid to upper crust (30-0 km depth) shows more lateral variation in density. This reflects the contrast between volcano-sedimentary domains and the felsic plutons that have lower densities (Figs. 3B, 4D).

### Geometry of the Abitibi Moho

The Moho depth derived from our 3D density model ranges from 32 to 45 km, with a mean depth of 39.7 km. A map of the Moho topography is shown in Fig. 4A. Large felsic intrusive complexes that have been exposed through exhumation are spatially correlated with a deep Moho. These plutons formed by buoyant felsic melt pooling within the crust through both partial crustal melting of the crust and melt fractionation[19] (Fig. 4D). Zones of anomalously shallow Moho are located between these domains and correlate with the locations at surface where base metal deposits are abundant (Figs. 4B and 5C: see "Discussion"). The location of rifting also correlates with the contrasting densities of the volcano-sedimentary domains and the felsic plutons (Figs. 3B, 4D), as noted in other Archean terranes[45]. The abundance of felsic plutons located at the flanks of the rift zones may reflect deeper levels of erosion at the rift flanks compared to the rift

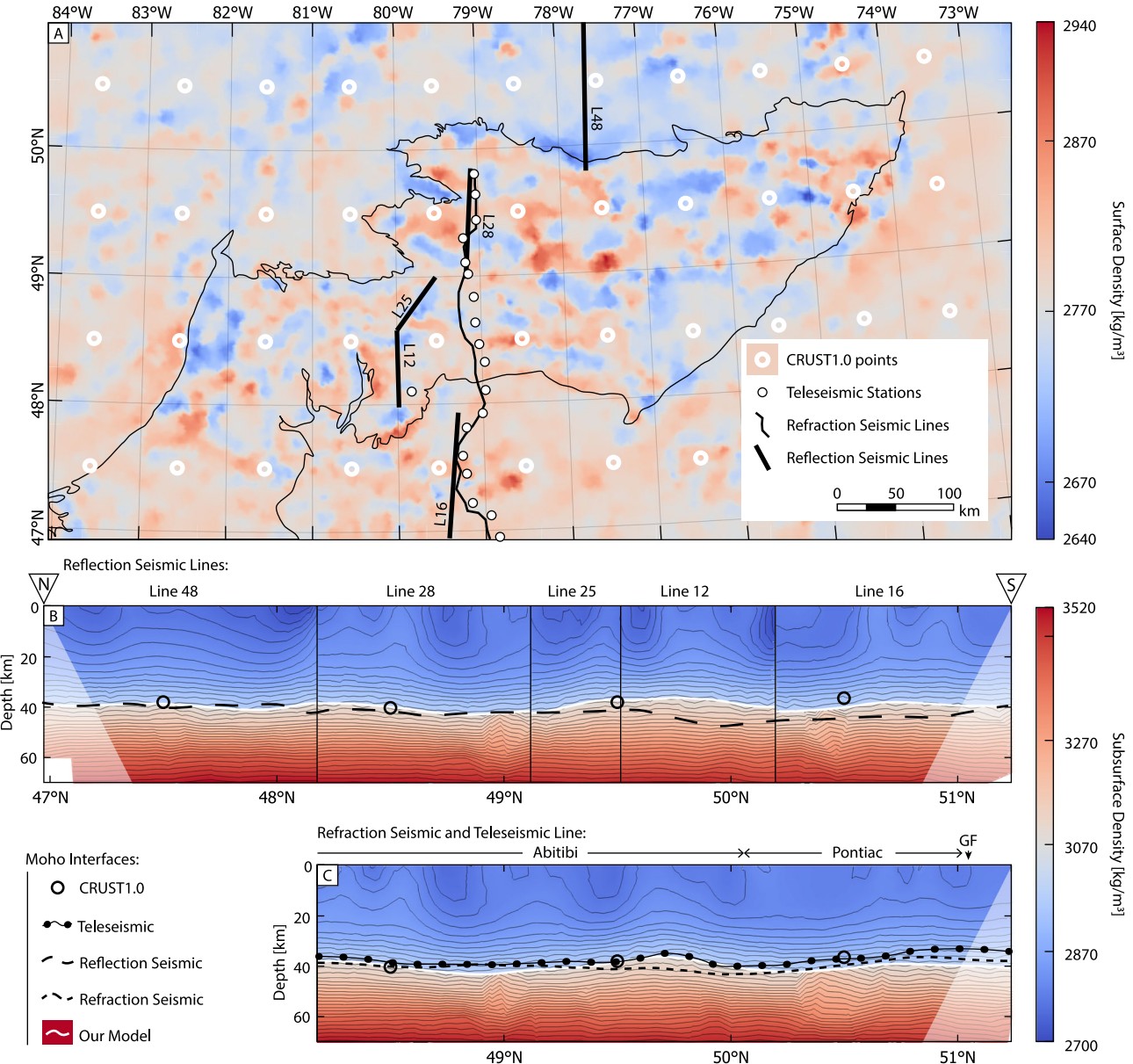

**Fig. 3 | A comparison of the three-dimensional density model of the Abitibi greenstone belt with previously published Moho depth models.** The map-view of the density model is shown in **A** and cross-sectional views in **B**, **C**. The Moho interface derived from the density model is compared to the Moho derived from the CRUST1.0 global model[43], a teleseismic model[44], five of the Lithoprobe reflection seismic lines[40], and a Lithoprobe seismic refraction model profile[41,42], as shown in **B**, **C**. In **A** the Abitibi is outlined in black[70].

valleys, the latter subsiding and being filled with denser mafic volcanics. Subsequent erosion would have led to a greater exposure of the pluton infused Neoarchean crust between the rifts resulting in the pattern we observe today[46,47].

The main E-W shallow Moho anomalies are in the north (Northern Volcanic Zone, NVZ, between Detour and Chibougamau), the south (Southern Volcanic Zone, SVZ, between Timmins and Amos), and in the southeast corner of the mapped area (Swayze area) (Fig. 5B). In the NVZ, the main rift is occupied by volcanic rocks having ages of 2.735–2.724 Ga (Deloro assemblage[19]). Here, three semi-continuous Moho anomalies are aligned east-west. The anomaly passing though Detour is 60 km in length, the Matagami anomaly is 100 km in length, and the Chibougamau anomaly is 120 km in length (see Figs. 1 and 5B for location names). In the SVZ, the main rift is occupied by volcanic rocks of 2.72–2.695 Ga (Kidd-Munro, Tisdale, and Blake River assemblages[19]). Five anomalies pass through the areas of Timmins,

Matheson, Noranda, Amos, and Val d'Or, including the 65 km long Timmins anomaly, the 65 km Matheson anomaly, the 125 km Noranda anomaly, the 60 km Amos anomaly, and the 60 km Val-d'Or anomaly. One oblique SW-NE trending corridor of shallow Moho also extends from Amos to Desmaraisville, following a portion of the 2.72–2.711 Kidd-Munro assemblage (Fig. 5B). This feature is approximately 130 km in length and 50 km wide.

Although regional north-south shortening has likely affected the overall geometry, the lengths and widths of the inferred rift zones (between 30 and 50 km in width between 60 and 130 km in length) are comparable to modern continental (e.g., East African Rift[48]) and oceanic rifts (e.g., the Lau Basin[25]). At the same time, we expect the relative topography of the Moho to be largely preserved through the deformation owing to the ductile behavior at depth. The lower Archean crust has been shown to flow under stress[49], a condition that leads to the thickening of the lower crust when undergoing post-rifting

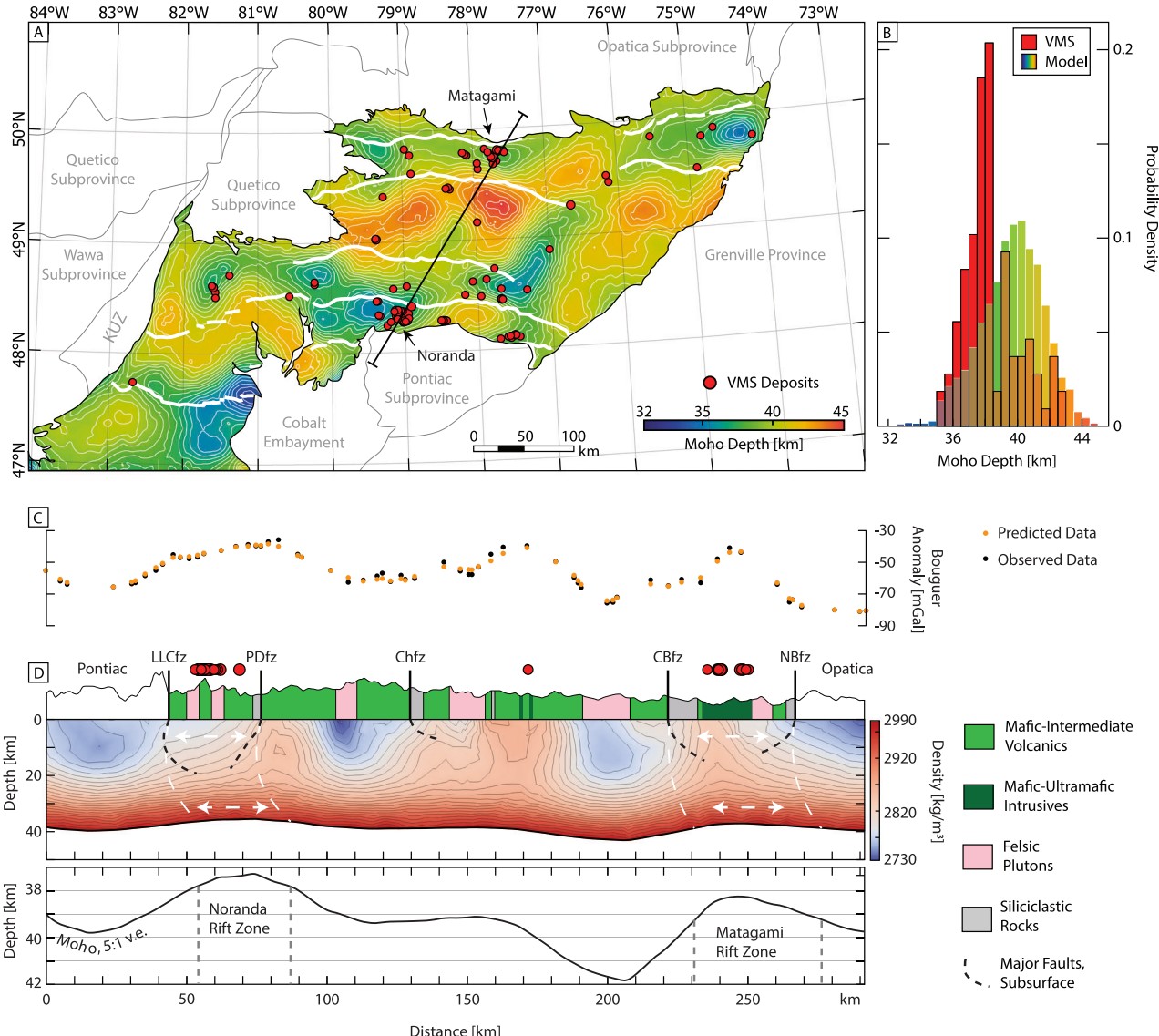

**Fig. 4 | A Moho depth map of the Abitibi greenstone belt, derived from the three-dimensional density model in this study. A** The depth to the Moho, contoured at 500 m intervals. Major faults and VMS deposit locations are overlain on the map. **B** A histogram of the Moho depth based on a 5 km grid spacing and the Moho depth below the known VMS deposits. **C** The observed and predicted processed Bouguer anomaly data along the profile in **A**. **D** A cross-section through the density model along the line shown in **A**. A shallowing of the Moho below the Noranda and Matagami VMS districts is evident. The subsurface fault geometry is from the Lithoprobe sections[40] and magnetotelluric modeling[71]. The lower panel in **D** is a 5:1 vertically exaggerated plot of the Moho topography, that highlights the variations. LLCfz Larder Lake-Cadillac fault zone, PDfz Porcupine-Dester fault zone, Chfz Chicobi fault zone, CBfz Casa Berardi fault zone, NBfz Northern Boundary fault zone.

compression. However, the shallowing of the Moho is still preserved despite the potential relative post-rifting thickening of the lower crust with respect to the upper crust[50].

## Discussion

The presented geophysical, geologic, and geochemical evidence best supports an extensional model of the Abitibi where the linear zones of shallow Moho are the root zones of relict Archean rifts. The surface manifestation of the rifts are zones of primarily mafic-intermediate volcanics that overlie a shallow Moho and contain abundant evidence of rift-related magmatic and hydrothermal activity. The intervening regions of crust that are infused by plutons are characterized by a deeper Moho and could, in the case of the Central Abitibi Microplate (Fig. 5A), represent a basement of older crust (e.g., Opatica crust[20]).

The boundaries of the Moho anomalies are marked by major crustal faults that are thought to have originated from initial graben-like extension in the Abitibi. They were later reactivated during the <2.69 Ga regional deformation[46,51]. The faults do not extend to the Moho but become listric at depths below approximately 20 km, coinciding with the brittle-ductile transition[52]. As a result, there is an offset between the faults at surface and the Moho topography (Fig. 4D). One interpretation is that the offset is a result of the later north-south compression, which led to the imbrication of the upper and mid crust and the southward displacement of the upper crust with respect to the lower crust[40]. Another interpretation is the presence of a lateral crustal flow through the Abitibi, which would have caused a displacement between the upper and lower crust, as has been observed in the Yilgarn Craton[49]. Our model suggests an ~10 km southward displacement of the upper crust relative to the lower crust (Fig. 4D).

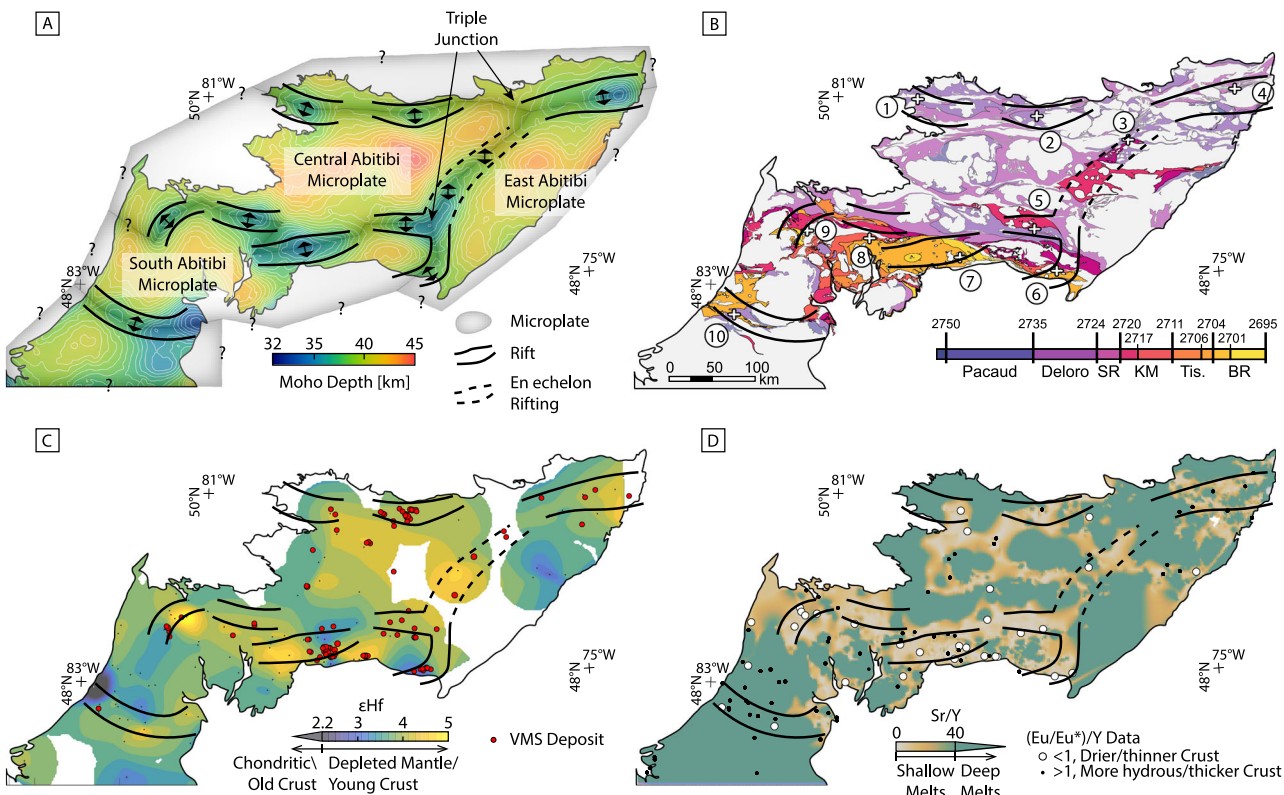

**Fig. 5 | The proposed rift model of the Abitibi compared to geologic and geochemical whole-rock and isotopic data. A** The Moho depth map with the corresponding inferred rifts and the boundaries of three potential Archean microplates. The black lines in each plot are the rift boundaries at depth and are projected to the surface in B-D. **B** A map of the lithostratigraphic assemblages of the Abitibi colored by age[70]. Towns and mining districts are numbered (see Fig. 1). **C** Schematic illustration of the Abitibi rift network from **A** showing the relationship to the locations of the known VMS deposits and overlain on an interpolated map of εHf data[20] of the igneous rocks (see text for discussion). **D** The interpreted Abitibi rift network superimposed on an interpolated map of Sr/Y data and (Eu/Eu\*)/Y data of the igneous rocks[20] (see "Discussion"). SR Stoughton-Roquemaure, KM Kidd-Munro, Tis. Tisdale, BR Blake River.

## Existence of triple junctions

A complex network of rifts in the Abitibi should have resulted in the formation of numerous triple junctions where the rift zones intersect, however, such structures have not been previously recognized. Triple junctions form at the boundary of three tectonic plates and are necessary to accommodate the expansion of a multi-plate system[6,53]. The connectivity of the rifts in our model is possible evidence of a Neoarchean microplate network. The modeled Moho topography is consistent with primarily E-W trending extensional zones and at least one oblique NE-SW zone (Fig. 5). The relative ages of the supracrustal rocks confirm that rifting migrated from north to south during expansion of the basins[54] (Fig. 5B). The NE-SW zone is occupied by volcanic rocks of varying ages, 2.72–2.711 Ga[19]. One interpretation is that it represents an extensional transform with overlapping en echelon volcanic systems between the early NVZ rifts and the younger SVZ rifts[20,21] (Fig. 5B). A modern analogue is the Central Volcanic Zone of Iceland, which hosts a series of en echelon volcanic systems that define a microplate mosaic between the Western and Eastern Volcanic Zones[55]. A similar feature in the Abitibi could have acted to transfer rifting from the NVZ to the SVZ. Oblique extension in this zone may also have limited the extent of crustal-scale faulting that otherwise dominated the rest of the terrane. Restoring the regional N-S compression and E-W shearing would clarify the kinematic history. The proposed microplate network requires the existence of at least two triple junctions in the Abitibi, located at opposite ends of the interpreted NE-SW extensional zone (Fig. 5A). Triple junctions are previously unrecognized features in the Abitibi, and their absence has been argued as evidence against Archean rifting[4].

## Rift-related mineral deposits and isotopic data

We show the inferred rift zones host numerous VMS deposits in clusters aligned along the Moho topographic anomalies (Fig. 5C), further supporting the interpretation that the Moho anomalies were once part of a submarine rift system with extensive hydrothermal activity[54]. In submarine rift environments, the shallowing of the Moho creates focused crustal-scale heat flow that drives melting, hydrothermal convection, and the formation of ore deposits[56]. The occurrence of VMS deposits, in particular, is widely associated with the submarine rift system, in both the modern and ancient rock record[57]. Figures 4A,C and 5 show the strong spatial correlation between the proposed zones of rifting and the occurrence of VMS deposits. The cross-sections through the NVZ and SVZ show notable shallowing of the Moho below two of the largest clusters of VMS deposits at Noranda and Matagami. Between these zones the Moho is generally deeper with sparse VMS occurrences. VMS deposits in the main rift zones are associated with primitive tholeiitic volcanic suites typical of melting at shallow depth[58]. In contrast, the deposits in the intervening areas are associated with calc-alkaline to alkaline felsic volcanic rocks formed by the rifting of thicker crust[59]. Trace element and isotopic data for the volcanic rocks support the relationship to crustal thickness. For example, in the middle of the Central Abitibi Microplate (Eu/Eu\*)/Y values suggest thicker crust with high La/Sm ratios indicating the presence of fractionated melts[20]. In contrast, the interpreted rift zones lie generally below crust with a more depleted mantle source indicated by εHf values (Fig. 5C), thinner crust (i.e., (Eu/Eu\*)/Y\*10,000 ratios <1; Fig. 5D)[20], more shallow melts (low Sr/Y ratios; Fig. 5D), and less fractionated melts (low La/Sm ratios[20]), all consistent with rifting of the

Archean crust[20,21]. The time-resolved εHf map, in particular, shows the existence of a rift-like feature at >2695 Ma that corresponds closely to the Moho topography shown in Fig. 5.

## Origin of rifting

Rifting in a modern plate tectonic context is linked to extensional forces driven by plate subduction, whereas in the Meso- to early Neoarchean other processes may have been more important, ranging from sagduction/drip tectonics[60,61] to plume-driven processes[5–7]. The results presented here provide evidence of at least local mobile-lid tectonics where rifting was focused along discrete corridors, whereas other tectonic regimes lead to disseminated extension across large areas[62]. The Moho topography of the Abitibi best fits a mobile-lid tectonic regime with distinct corridors of shallow Moho encircling regions of thicker crust (Fig. 5). Numerical modeling and other observations[6,63,64] have shown that mantle plumes could have initiated rifting in hot Archean crust. In a homogeneous crust, a plume may generate many rift centers with transform zones accommodating the extension between the rifts, as in Iceland. This may be analogous to the Abitibi rifting, although the model presented here does not resolve the far-field mechanism behind rifting in the Abitibi.

This study presents a gravity-derived model of the Moho surface beneath the Abitibi greenstone belt that reveals a possible Neoarchean rift network. The geometry of the proposed microplate mosaic is supported by trace element and isotopic data of associated volcanic rocks and the occurrence of rift-related mineral deposits. The results show a strong empirical relationship between mantle plume activity, microplate formation, and major mineral systems in a local mobile-lid tectonic environment.

## Methods

Our models were constructed by minimum-structure style inversion modeling[65], a predictive method used to produce a model of the subsurface that generates a geophysical signal closely resembling that measured through surveying. In our study we modeled gravity data to produce a density model of the subsurface. The model was parameterized by discretizing the sub-surface with tetrahedral cells that can accurately fit the topographic surface while also varying in size to reflect the decreasing model sensitivity with depth. Figure S1 shows the tetrahedral mesh, which was constructed using the program *TetGen*[66] and used to discretize the Abitibi crust and upper mantle. The model contained 642,516 cells.

### Objective function structure

Inversion modeling is largely a minimization process, where a term known as an objective function is defined and minimized through an iterative process. The inversion approach used considers an objective function structured as

$$\phi = \lambda\phi_d + \phi_m. \tag{1}$$

This objective function is composed of two primary components: the data misfit term, $\phi_d$, and the model measure term, $\phi_m$. The trade-off parameter $\lambda$ is present between the data misfit and model measure to allow the data misfit to achieve an appropriate target value during the minimization process[65]. The data misfit is composed as

$$\phi_d = \frac{1}{N}\sum_{i=1}^{N}\frac{\left(d_{i,pred} - d_{i,obs}\right)^2}{\sigma_i^2} \tag{2}$$

with $d_{i,pred}$ the predicted data (i.e., the forward signal from the inversion model), $d_{i,obs}$ the observed data that is being modeled, $\sigma_i$ the Gaussian noise of the observed data, and $N$ the number of data points. This function acts to minimize the difference between the observed

and predicted data, which results in an inversion model that produces a forward signal closely resembling the observed data. During the modeling process we wish to avoid fitting the observed data's noise, which is why $\sigma_i$ is present in the denominator, as this allows the expected value of the data misfit to converge to $N$. Normalizing the misfit by $N$ then leads to the term to minimize to a value of 1. The predicted data, $d_{i,pred}$, calculated from the model, are calculated using Okabe's forward calculation algorithm[67].

Simply minimizing the data misfit through inverse modeling would lead to a non-unique solution. To fix the non-uniqueness issue a model measure is added to the objective function, which acts to smooth density variations throughout the cell-based model. This model smoothing implies a sort of "Occam's Razor" approach to the inverse modeling, where when presented with a large number of equally valid model solutions, the simplest model is chosen as the best[65]. We employ the classic L2 norm smoothing to the model measure,

$$\phi_m = \boldsymbol{v}_f^T \boldsymbol{W}_s \left(\boldsymbol{D}_f \boldsymbol{m}\right)^2. \tag{3}$$

Within the above model measure, $\boldsymbol{W}_s$ is a diagonal weighting matrix, $\boldsymbol{v}_f$ an array of integration cell volumes, and $\boldsymbol{D}_f$ is a general difference matrix applied to the array of model parameters $\boldsymbol{m}$[68]. A variation of Li and Oldenburg's sensitivity weighting was employed to preferentially weight cells with low sensitivities[69], as otherwise the inverse modeling would preferentially place all density anomalies near-surface. The sensitivity weighting matrix was composed of elements

$$w_j = \left(\sum_{i=1}^{M}\left|\frac{G_{ij}}{v_j}\right|^2\right)^{1/2}, \tag{4}$$

for the $j^{th}$ cell in the mesh with $\boldsymbol{M}$ cells, whose volume is $v_j$. $\boldsymbol{G}$ denotes the inverse problem's sensitivity matrix for the gravity problem, with elements $G_{ij}$.

Often the difference matrix, $\boldsymbol{D}_f$, is left in its general form that calculates the difference in physical properties between each cell and its direct neighbours (i.e., the smoothing term). However, model smoothing can be further controlled along each of the three Cartesian directions by defining the difference matrix as

$$\boldsymbol{D}_f = \omega_x\boldsymbol{D}_x + \omega_y\boldsymbol{D}_y + \omega_z\boldsymbol{D}_z, \tag{5}$$

with $\boldsymbol{D}_x, \boldsymbol{D}_y, \boldsymbol{D}_z$ representing gradient operators in each Cartesian direction, and $\omega_x, \omega_y, \omega_z$ are scalar weights[68]. By lowering one of the Cartesian weights, a model can be produced with more discrete changes in its physical property in that direction. $\omega_x, \omega_y, \omega_z = 1$ represents the scenario where there is equal smoothing in all directions. We used a $\omega_z = 10^{-5}$ to promote sharper vertical variations in density, as at large scales the Earth's subsurface is primarily composed of a generally layered system of increasing density with depth.

### Reference model

The Bouguer dataset was taken from the Natural Resources Canada database, which is a compiled dataset of surface-level gravity readings[31]. Inverting the data as is resulted in negative relative density features at depth (relative to 2670 kg/m³) that should have corresponded to the upper mantle, and no density gradient with depth as would be expected (Fig. S2A). This discrepancy in baseline densities results from the latitude correction performed on the free-air gravity data, which removes an average signal of the assumed elliptical Earth model from the data. After the latitude correction, any anomalous density bodies in the crust will have densities that are relative to that of the crust, but any anomalous bodies in the mantle will have densities

relative to the mantle. A reference model of the crust was needed to remove the discrepancy in crustal versus mantle density anomalies. An optimized reference model of relative densities equal to 0.0106*z kg/m$^3$ was used (*z* being depth). This reference model was derived by iteratively increasing the density gradient of the model, adding the forward signal of the reference model to the Bouguer anomaly data, inverting the data and comparing the geometry of the 3070 kg/m$^3$ iso-density contour to the Moho from modelled seismic sections[40,42,44]. This resulting forward signal of the reference model was 1092 mGal. Inverting the Bouguer anomaly data with the forward signal of the reference model resulted in the model shown in Fig. S2B.

### A priori moho structure constraints

In addition to Archean crust being thicker than modern day seafloor crust, the Abitibi Moho is found to be a sharper boundary in sound velocity and density[37]. Considering that density distributions change smoothly with depth, as in our minimum-structure type inversion modeling[65], the Moho under continental crust can be represented by the approximate 3070 kg/m$^3$ iso-density layer[26]. After the initial inversion model was produced, the 3070 kg/m$^3$ iso-density layer was extracted and used to improve the model of the Abitibi. It was improved by adding constraints to the density values that the model could resolve in the portion of the model that corresponded to the crust versus the lower portion defined as mantle. This two-layer model was made by including the extracted 3070 kg/m$^3$ iso-density interface into the mesh design for the second model (Fig. S2C). The density constraints of the two layers were: 1) between 2600 and 3000 kg/m$^3$ in the crust[38], and 2) between 3070 kg/m$^3$ and infinity for the mantle. This final version of the model then had a discrete change in density across the Moho.

### Forward model verification

To verify that the density anomalies in the upper mantle shaped the topography of the modeled Moho, we created a series of forward models of components of our inversion model (Fig. S3). The first forward model calculated the gravity signal due to anomalies above the Moho, i.e., crustal anomalies (Fig. S3E). The second forward model calculated the gravity signal from the Moho, treating it as representing the 3070 kg/m$^3$ interface as it was in the inversion model (Fig. S3C). A third forward model calculated the gravity signal from the component of the inversion model below the Moho, from which the forward signal of the Moho (Fig. S3C) was subtracted. The remaining gravity signal represented density variations in the upper mantle (Fig. S3D). Comparing the forward signals of the crustal anomalies and Moho anomalies to the modeled data (Fig. S3A) it is possible to quantify how much of the modeled data was fit due to density features in the crust versus features in the upper mantle. As expected, most of the data was fit by crustal density anomalies, but a significant, more regional component of the data had to be fit by the deep Moho density feature. The amplitude of the forward signal of the Moho component was ~20–30 mGal (Fig. S3D), much greater than either the Model Data Residual (Fig. S3B) or the data uncertainty that averaged 0.3 mGal[31]. The amplitudes of the forward signal from the crust were slightly higher, at 20–50 mGal, but occurred over shorter wavelengths. An example section is shown in Fig. S3F, which shows the amplitudes of the modeled data, the forward signal from the crust, Moho, and upper mantle, and the model data residual. All the data along this section have been shifted relative to their minimum value. The section shows that although density variations in the upper mantle did contribute in fitting the modeled data, their amplitude was small and in some cases less than the amplitude of the model's data residual.

### Data availability

Gravity and topography data used in the study can be found through the Government of Canada's geophysical data portal (http://gdr.agg.nrcan.gc.ca/gdrdap/dap/index-eng.php?db_project=10013).

### Code availability

The inversion software used, *Vidi*, is property of Memorial University of Newfoundland and Geotexera Inc., and is available upon request for academic purposes only. Please contact Dr. Lelièvre for further details (plelievre@mta.ca).

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

## Acknowledgements

We would like to thank Howard Poulsen for comments that improved an early version of this paper. All inversion modeling was performed on Digital Research Alliance of Canada clusters, which are funded by the Government of Canada. This project was funded through the Canada First Research Excellence Fund (CFREF, Metal Earth). The Natural Sciences and Engineering Research Council of Canada (NSERC) and CFREF are acknowledged for support of this work through research grants and project funding to the M.H. This is Metal Earth contribution MERC-ME-2025-04.

## Author contributions
C.G. and M.H. wrote the initial paper and, along with E.B. and A.B., designed the study. The geophysical modelling was performed by C.G. and P.L. All authors discussed the results, provided feedback, and contributed to the final paper.

## Competing interests
The authors declare no competing interests.
