## [Transparent Peer Review file · Nature Communications]

Archean rifts and triple-junctions revealed by gravity modeling of the southern Superior Craton

Corresponding Author: Dr Christopher Galley

Version 0:

Reviewer comments:

Reviewer #1

(Remarks to the Author)

Thank you for submitting this work. The results of the geophysics were extremely interesting and for the first time directly image Archean rift architecture and systems. However, there are some aspects that need to be improved in my view - see my detailed comments for revisions. Good luck with the revisions.

Reviewer #2

(Remarks to the Author)

This is an interesting paper suggesting development of variably oriented rift centers organized into triple junction(s) in the Archean Superior Craton. This suggestion is supported by analyses of gravity data combined with seismic profiles. It is suggested that (multi-directional) extension and subduction were induced by Archean mantle plumes, which mark transition from stagnant to more mobile lid behavior. This new data and hypothesis are potentially important for better understanding of the Archean geodynamics. However, materials in support of this hypothesis are not very well presented. Many geographical and geological features discussed in the text are not indicated in figures. Conceptual sketch representing the geodynamic hypothesis and the proposed geometry of micro-plates, rift centers and triple junction(s) is absent. This makes argumentation discussed in the paper difficult to follow. Specific comments are given below.

Taras Gerya, Zurich, 28.11.2024

Specific comments

Fig. 4 Caption. "in A) showing a shallowing of the Moho below the Noranda and Matagami VMS districts." These districts are not indicated on the figure.

Line 139. "We propose the linear zones of shallow Moho are relict Archean rifts. The interpreted rifts are environments of primarily mafic-intermediate volcanics that overlie a shallow Moho. The pluton-infused regions are characterized by a deeper Moho and could represent early continental crust and/or a basement of older Opatica crust 11."

These the linear zones of shallow Moho need to be clearly indicated in Fig.4A, which is otherwise difficult to interpret. Geometry of the proposed triple rift junction needs to be outlined.

Line 160. "Abitibi is consistent with primarily E-W trending extensional zones and at least one oblique NE-SW trending zone (Fig. 4A). The main E-W anomalies are in the Northern Volcanic Zone (NVZ; between Detour and Chibougamau), the Southern Volcanic Zone (SVZ; between Timmins and Amos), and in the southeast corner of the mapped area (Swayze area) (Fig. 1). In the NVZ, the main rift is occupied by volcanic rocks of 2.735-2.724 Ga age (Deloro assemblage10). The Detour anomaly is 60 km in length, the Matagami anomaly is 100 km in length, and the Chibougamau anomaly is 120 km in length. In the SVZ, the main rift is occupied by volcanic rocks of 2.72-2.695 Ga (Kidd-Munro, Tisdale, and Blake River assemblages10)."

All these names are not in Fig.4A and it is difficult to guess which features are referred.

Line 207. "Bouguer gravity data collected over the Abitibi Greenstone Belt was modelled to study the architecture of the Neoproterozoic crust. The model suggests the Abitibi greenstone belt was composed of a number of pluton-infused micro-plates surrounded by Neoproterozoic rift centers. The model also suggests the existence of triple junctions, a previously unrecognized feature whose absence in the rock record has been called upon to dismiss the presence of Archean rifting 37." This discussion is not supported by an adequate conceptual figure. This could be a separate Figure 5 containing the interpreted sketch based on Fig. 4A with clearly indicated micro-plates, rift centers and triple junction(s).

Reviewer #3

(Remarks to the Author)

Version 1:

Reviewer comments:

Reviewer #1

(Remarks to the Author)

Nice paper - thanks for those revisions. Some more work to do, but the ms is close. I think the revisions are fairly straightforward, but may take a bit of time to get really polished. More detailed comments are in the report attached.

Reviewer #2

(Remarks to the Author)

The authors made appropriate revisions to the paper, which is now suitable for publication.
Taras Gerya, Zurich, 17.02.2025

Reviewer #3

(Remarks to the Author)

Version 2:

Reviewer comments:

Reviewer #1

(Remarks to the Author)

This is the third time I have reviewed this paper, and I am now happy for the paper to be accepted. Thank you to the authors for making the corrections in good faith. I think this will be an interesting contribution.

However, as I said in my initial review, this is based on the assumption that a geophysicist has reviewed and cleared this manuscript, as I do not have the expertise to check that element of the paper.

A few final comments:

Line 30-31: remove somehow, add 'potentially'

Line 162: This should be 'deeper' level of erosion - if these areas represent the 'horsts' then they eroded more deeply to reveal the intrusive rocks (as they were at a higher level), whereas the grabens or basins still contain the volcanic sequences. Line 163 can still be true in this scenario. However, what you say in line 164 suggests you already know this, so perhaps just changing the 'levels of erosion' to 'levels of subsidence' is more accurate.

Line 297-299: Nice way to finish!

Reviewer #3

(Remarks to the Author)

Reviewer #4

(Remarks to the Author)

Please note, that is not a complete review of the manuscript, but some thoughts on the gravity modelling performed as part of the study and the concerns by one reviewer.

As such, the modelling seems to be correct, but in my view is explained in unnecessary complicated matter.

1) Processed Bouguer anomaly

The processed Bouguer anomaly is simply a shifted Bouguer anomaly by adding a constant. Hence, I would recommend to show the Bouguer anomaly instead of the processed one.

2) The shift has implications for modelling. Figure S3 illustrates that by showing that both models have the same gravity anomaly disregarding the shift (and I would omit this figure). Such a shift is often treated manually, by an average value or a reference model depending on the application. Here, a reference model would be most appropriate and this is sort of expressed by a 65 km thick slab of 400 kg/m^3 (and I suggest that authors use SI units for density). This shift in the inversion is somewhat similar to setting a reference depth in methods like Parker-Oldenburg.

3) This is also seen in the resulting density maps in Figure S2 (in which depth I wonder), which for result A shows that simple depth weighting is not sufficient. Only when the anomalies are shifted to positive values the inversion provides increasing densities with depth.

4) Coming to densities. A contrast of 400 kg/m^3 does not indicate a mantle density of 3070 kg/m^3 as it is well known that 2670 kg/m^3 is an okay value for near-surface density, but that density and velocity increase with depth (e.g. Zoback & Mooney 2002) and that there is a jump at the Moho. I have the feeling and this can't be seen from your colour scale, that you smear out the contrast (as densities in the mantle continue to increase). That would be fine, if you demonstrate that the model response is the same, when all densities >3.07 are put to a constant value of 0.2-0.4 larger than the lower crust.

5) "We opted not to use the surface-based modelling method, instead using the more sophisticated, albeit more computationally demanding, minimum-structure inversion method (see Methods for details), to account for the known heterogeneity in the density of the Abitibi crust"

This sentence is a bit a summary of the authors opinion, but I would argue that the word sophisticated is stretching it a bit. Yes, inverting for the density change with depth is okay and its advantage is that it provides crustal densities which appear reasonable. However, I would argue that you get the same result, when you A) filter the data, B) invert the long-wavelength part for the Moho (with reference depth of 35 km and density contrast of 400 kg/m^3), and C) invert the short-wavelength residual for crustal internal variations.

6) Furthermore, I disagree with Lines 381-388 and that you corrected for something done wrong in a normal correction. I applaud your honesty, that you state this correction as most modellers simply apply a shift, but this is also not a meaningless quantity, and we are simply speaking about a reference model, which also has isostatic implications.

7) The inversion itself is well described, but the setting of the slab is actually the main factor for your results. I had to carefully think about what actually has been done and can completely understand the concerns and confusion of one of the reviewers.

In summary, I would omit Figure S3, put some of the technical descriptions on the inversion in the supplement, but discuss the role of the slab or reference model in the main text as this is the most crucial factor for the resulting models and refer to a normal use of anomalies. The processed Bouguer anomaly creates an unnecessary level of confusion and does not contain any new information. For gravity experts, shifts and reference models are nothing new. For non-experts that simply mystifies the gravity modelling.

Jörg Ebbing

Version 3:

Reviewer comments:

Reviewer #4

(Remarks to the Author)

The revised version has improved a lot as now the gravity inversion is explained in a clear and transparent way. One could still argue about the dis- and advantages of the inversion procedure compared to other approaches. As such the approach is sound and as with all potential field inversions, not one recipe is applicable to all areas. The authors admit this and reflect on this in the discussion adequately. The methods part describes all details for the expert reader, so that I recommend publication.

Reviewer # 1's comments

Reviewer 1 found the manuscript to provide important insight into Archean crustal and tectonic activity, with a dataset and imagery that were convincing and match other datasets related to the Abitibi crust. The reviewer was not convinced that this paper needed to get into "the Archean geodynamics controversy", and we agree. Rather, the impact of the paper is in imaging relict Archean rifts alone, and seemed less concerned about the plate tectonic debate. Nevertheless, we have added a fuller discussion of how alternative tectonic scenarios would influence the interpretation of our results. The abstract is re-written, and the paper has been edited to be less ambiguous about the general findings.

Major points

- 1. More figures to backup the arguments and models – at least one schematic figure clearly showing what they consider 'microplates'**

A new illustration has been included as Figure 5.

- 2. A figure showing how they see the temporal evolution of the Abitibi, in regard to their model/data, including the late multi-phase compression.**

We have further discussed the impact of the late compression event and show the age progression of crustal growth in our new synoptic Figure 5B.

- 3. More links to the isotopic maps of Mole et al (2021) to reinforce the arguments.**

We agree and have discussed the similarities to Mole et al. more in the revised text and in Figure 5.

- 4. Are microplates required to explain your data? The Reviewer called for more justification of a model whereby the proposed complex network of plates and triple junctions is really a microplate mosaic, citing examples from Iceland, Lau Basin, Japan.**

Different views have been expressed about what constitutes a "microplate mosaic", ranging from subduction driven fragmentation to plume induced fragmentation. To clarify, we have included a straightforward definition (i.e., crustal fragments that are encircled by rifts) that do not imply a particular tectonic regime. In particular, proximity to subduction is not necessary for microplate formation. We cite the key literature on this topic (Mammerickx & Klitgord, 1982; Mueller et al., 2017) that describes the typical sizes of microplates and their kinematics that closely resemble the structures inferred from our model.

- 5. You have 41 refs, but Nat comms allows up to 70, hence I have added a number of suggestions.**

The suggested references have been added, and the text and figures have been revised accordingly.

- 6. Include more on the geochemical and isotopic mapping that has been done, which supports your arguments.**

We added more references to the extensive work of Mole et al. to strengthen the interpretations of the geophysical model results.

- 7. The reviewer did not see why rift networks and triple junctions could not occur in a stagnant lid or "non-plate" scenario, citing additional details in the comments below.**

This is addressed in comment #17 below and in the revised manuscript.

- 8. The reviewer was not sure this paper needs to get into the Archean geodynamics controversy – the data on rifts is of high impact enough. From what I can see, it could be used to make arguments for multiple settings, and does not support the existence of plate tectonics, perse.**

Our model focuses on physical evidence in the deep crust and mantle for the existence of Archean rifts in the Abitibi, and how they could have been connected. The geometry and potential connectivity of the rifts point to a microplate model, and we describe how this could apply to a variety of tectonic processes from plate-dominated to non-plate dominated, including a plume setting.

- 9. In the end, the reviewer thought it would be good to discuss various settings, AFTER presenting the main finding of the rift architecture, including what might be the best modern analogy.**

We added a new section specifically addressing tectonic scenarios for the modelled rift zones, including specific comparisons to the microplate architecture of present-day Iceland as an example. We argue the Moho topography of the Abitibi best fits a horizontal tectonic regime, which contrasts with the coupled shortening and extension of the crust that would be associated with diapirism and a vertical tectonic regime.

- 10. Use of geology to support the model (e.g., the occurrence of komatiite).**

Many known occurrences of komatiite align with the rifts, but some don't. This is partly because the modelled rifts are deep features, and komatiites, like the major faults, are near-surface features that may be displaced

relative to the Moho topography due to lateral crustal flow (as seen in the Yilgarn). This is discussed in the revised text, but a more detailed description of the surface geology, vis-à-vis the rift model, is being prepared for another manuscript.

11. There is good evidence that these rifts were continental in nature (syn-volcanic zircon from these rifts do not suggest an oceanic origin, as well as rare, but constrained Mesoarchean zircon inheritance). How does that affect the model?

See response to comment #4.

12. Can you infer plate tectonic scale features based on the scale of features you have mapped – i.e., in a diagram that summarises your findings and clarifies the scaling.

The new Figure 5 clarifies these relationships and the scales. We describe in the revised text (and in the response to comment #17) our interpretation of the rift geometry with comparisons to different tectonic settings.

13. Line 10: Whether plate tectonics was indeed occurring in the late Archean is still debated – it should not be presented as fact.

We have adjusted the language accordingly.

14. Line 27: This model has been published by others for the Superior and other cratons and should be cited.

We have added the references.

15. Line 28: Weak hot crust would be more likely to flow laterally, which could possibly aid rifting.

We have clarified this statement, emphasizing the comparison of hot Archean crust to modern seafloor spreading centers and subduction-driven rifting.

16. Line 32: Again, be careful stating debated points as facts.

We have changed the sentence to emphasize the general consensus, rather than a statement of fact.

17. Line 36: Rifts do not necessarily = mobile lid.

See response to comment #4. We have clarified the distinction between horizontal (or mobile-lid) and vertical tectonics, as objectively as possible.

In horizontal tectonics the crust comprises many rigid or semi-rigid lithospheric plates bounded by divergent, convergent, and/or transform margins. We believe this is what our model shows and emphasize that this is the extent of our results, without reference to the motion of the plates (only the extension and fragmentation). We think most readers already recognize that the broader aspects of subduction are still a matter of debate, and we have tried to characterize them as such.

18. Line 37: Previous work on Abitibi rifting.

We have revised the text to more fully describe the previous work on Abitibi rifting, based on isotopic mapping, and agree it is great that these two completely independent datasets are showing such similar features.

19. Line 50: What is the evidence that the Moho topography have survived 2.6 Ga of modification?

We have added a comparison of our model to the time-resolved isotope data from Mole et al. (2022) and also to the distribution of VMS, as two independent pieces of evidence for survival of the Archean Moho topography and the corresponding rifts, as suggested by the reviewer. The time-resolved ϵ_{Hf} map, in particular, shows the existence of a rift-like feature at >2695 Ma that corresponds closely to the Moho topography shown in Figure 5.

20. Line 62: Why is the mid-lower crust not affected by the deformation?

We explain in the revised manuscript that the N-S shortening thickened the crust by upper crustal imbrication above the brittle-ductile transition zone. Seismic profiles have shown that this imbrication does not extend to the Moho, thus the Archean Moho topography is preserved. The mid-lower crust is also not affected because is it largely below the BDZ.

21. Line 83. Is it possible the intrusive (less dense) and volcanic areas (more dense) are creating the appearance of belts of deep/shallow Moho? Dense volcanic rocks are understandably more abundant in rifts, but why would intrusives dominate the flanks?

Deeper features, such as Moho topography, which we interpret as rifts, produce long wavelength, low amplitude anomalies in the gravity data. Near-surface features produce shorter wavelength, higher amplitude anomalies. The iterative process of the inversion modelling is able to resolve these in 3D. We explain in the revised text (see also comment #24) that the abundance of felsic plutons at the flanks of the rift zones may

reflect shallower levels of erosion at the rift flanks compared to the rift valleys, the latter subsiding and being filled with more dense mafic volcanics. Subsequent erosion would also have led to a greater exposure of the pluton infused crust between the rifts resulting in the pattern we observe today, acknowledging that some intrusions may be younger than the rift system.

22. Line 93: Why do you not show any 3D images?

The model is 3D, but we find it is more informative to show aspects of the model in plan view or cross sections (Fig. 3B and 4D) rather than in a 3D perspective. We could extract a 3D surface from the model to represent the Moho, but studying it is best done in section and elevation/depth maps.

23. Line 94: I would argue here that you have added further resolution, and confirmed the architecture, but it is not the first rift architecture map of the Abitibi.

We have revised to text throughout the manuscript to more accurately describe the contribution made here relative to the sparse Moho information available from seismic studies and isotope mapping.

24. Line 116: If the rifts are dominated by volcanics, perhaps this is because they were localised into grabens, which filled with volcanic material and form a lower crustal level. The plutons are represented by the felsic intrusion dominated flanks, which are horsts. There are likely felsic intrusions under the grabens, they just aren't exposed yet. This is important because many people think we cannot image the rifts as has been done here and isotopically.

We have added this discussion to the text and in response to comment #21.

25. Line 119: What about the thickness of volcanic rocks?

We have clarified this in the text and in response to comments #21 and 24.

26. Line 122: Can we presume these rifts would have been much wider, and what we are seeing is the inverted, closed versions? Also the 'depth' of the moho is presumably a minimum?

Yes, this is correct, and we have emphasized that the modelled features are post-tectonic. We have tried to restore the N-S compression and E-W

shearing but chose to leave that for another paper, given that it is not directly related to the gravity modelling. We also avoid the term “crustal thickness”, as opposed to “Moho depth” to avoid any assumptions about the depth of erosion and thickness of the crust at ~2.7Ga. Instead, we focus only on the topography of the Moho.

27. Line 139: Please clarify what crust you think is being rifted.

We note that the rifts are similar in their architecture to continental rifts and may have developed in crust with a thickness similar to that of modern continental crust. However, we do not discuss compositional or petrological implications that would be far beyond the scope of this paper.

28. Line 152: Lateral ‘flow’ of hot mafic crust could explain how the lower and mid-upper crust can become dissociated.

We have added this point to the discussion.

29. Line 158: Rifting versus discrete plates and triple junctions.

See response to comment #4 and #17. We have provided the accepted definition of microplates in the paper and discuss them several times in the context of crustal fragments that are encircled by rifts. We do not imply a particular tectonic regime and consider that they can form in either crustal type (oceanic or continental). We acknowledge that rifting can occur without discrete plates or triple junctions, but we show here that triple junctions likely existed, requiring a microplate architecture.

30. Line 160: Why are we only hearing about this oblique zone now?

We have moved the description of the SW-NE trending corridor of shallow Moho to the results to better introduce this feature in the context of the modelling.

31. Line 166: How continuous are these rifts?

We have described the segmentation of the rifts (and the scale of segmentation) which is similar to modern continental and ocean rifts, including the boundaries of microplates.

32. Line 168: It would be good to see the rifts presented as one continuous system.

We have illustrated the potential continuity of the rifts in the new synoptic Figure 5.

33. Line 171: Please reference points that are not original.

References have been provided where indicated.

34. Line 175: This is the interpretation of Mole et al 2021, 2022.

References have been provided where indicated.

35. Line 175: Note the occurrence of these kind of transform systems in Iceland.

We have added the comparison.

36. Line 180: Reference the isotopic mapping which imaged this feature and supports your identification of it.

We have added the comparison.

37. Line 180: You have excellent evidence for a complex rift system, but can you demonstrate the rifts separate 'plates'? Isotopic and geochronologic evidence suggests pre-existing crust, so how does this factor into your model? Triple junctions are usually between oceanic or mafic plates.

We have addressed this question in our definition of microplates, which are closed polygons encircled by rifts. Triple junctions are a requirement of microplate formation, regardless of the makeup (oceanic or continental) or age of the crust. The oblique shallow Moho anomaly in the east creates the possibility of triple junctions in our model. We have tried to re-emphasize these points in the revised text.

38. Line 180: If you are going to make the link to horizontal tectonics, here would be best.

This discussion has been moved to another location.

39. Line 183: The link to ore deposits requires more references, including in the introduction.

Done.

40. Line 195: Be more clear about what crust you think is being rifted (oceanic versus continental).

See responses to comments #4 and #29. We chose not to include a discussion of the makeup of the crust, but focus instead on the Moho topography as the main control on rifting.

41. Line 196: You have not discussed to any extend the nature of the crust.

See response to comment #40.

42. Line 204: Wording

Adjusted.

43. Line 198-204: You do not need to get into the Archean geodynamics controversy with this paper. The identification of the rift architecture is big enough news.

We have clarified that the rift model does not imply a particular geodynamic, including subduction. For detailed comments regarding the different scenarios see responses to #4 and 17.

44. Line 207: The end of the paper is not decisive about linking the data to plate tectonics and 'plate boundaries'.

We have modified the ending to better match the tone of the paper throughout.

45. Figure 2: Improvements

We have added the outline to the data figure, and the corresponding geographical details are provided in Fig. 1. We have added landmarks and labels to the other figures, including the new Fig. 5.

46. Figure 3: The red-blue colour profiles used are not generally good for colour-blind people and may not present data clearly. Also be consistent across all figures.

We are familiar with Cramer et al. 2020 and use the CET colour ramps the paper discusses. The red-white-blue colour ramp in Fig. 3 is standard for density figures, but the same bounds are not used in Fig. 4. Crust is mainly blue and the mantle red. This is explained in the caption. Variations in density are the main focus of Fig. 4, and particularly Fig. 4D, where we removed the mantle component to highlight just the variations in the crust.

47. We encourage the authors to ensure all raw data are available for others to repeat and test their model.

We have added the data and instructions on how to access the modelling program.

Reviewer # 2's comments

Reviewer #2 found the manuscript to contain important new data and hypotheses that potentially offer better understanding of Archean geodynamics. However, improvements are suggested for better presentation of the materials in support of

model, in particular several details of the figures and a conceptual sketch representing the proposed geometry of microplates, rift centers and triple junction(s).

Specific comments

- 1. Fig. 4. The districts are not indicated on the figure.**

We have added the labels to the figure.

- 2. Line 139. The linear zones of shallow Moho need to be clearly indicated in Fig.4A, which is otherwise difficult to interpret. Geometry of the proposed triple rift junction needs to be outlined.**

We have created a new Figure 5 to better outline the rifts, and compare them to the Moho depth model. We also introduce the role of oblique extension earlier in the paper that highlights the connectivity of the rifts and the locations of possible triple junctions that support the microplate model.

- 3. Line 160. All these names are not in Fig.4A and it is difficult to guess which features are being referred to.**

We have added all this information in the new Figure 5.

- 4. Line 207. This discussion is not supported by an adequate conceptual figure containing the interpreted sketch based of Fig. 4A, the indicated microplates, rift centers and triple junction(s).**

These features are all indicated in the new Figure 5.

Reviewer # 3's comments

- 1. Line 49. There is an assumption in this paper that studies of today's gravity field can be used to infer crustal structure in billion-year-old cratons. This assumption should be stated and justified.**

We have revised the text to better explain how the Archean Moho topography is preserved through subsequent deformation (see response to comment #20 of Reviewer #1). We outline in the introduction how gravity inversion modelling can reveal the original Archean Moho and that any subsequent deformation has not erased the original topography. We have also discussed how studies have shown the lower crust/ Moho is largely unobscured and can indeed support large stresses on long geological timescales (Benn, 2006).

- 2. Lines 68/69. The statement that "Inversion modelling creates a petrophysical model" is misleading. Inversion in gravity interpretation is a procedure that creates a body with a certain specified density contrast with its surroundings directly from an observed gravity anomaly.**

We have clarified this statement in the Methods section. There are two main types of inversion modelling parametrizations: geometric and voxel-based. In geometric inversion modelling, the physical properties in the model are fixed (assigned) and the geometry of the body is solved for. We perform minimum-structure inversion modelling, which uses a voxel-based parameterization. This type of inversion modelling produces a pixelized petrophysical model of the sub-surface.

- 3. Line 71/72. Why is it relevant that the vidi inversion code was run on Canada's narval cluster? Is the code limited to certain platforms? Is it publicly available?**

We mention the Narval cluster to document the infrastructure and resources used in the modelling. The code is not limited to certain platforms. We have added information on how the program can be accessed in the Code Availability section. It was developed at Memorial University of Newfoundland, and is now the main tool employed by Geotexera Inc., a commercial company. The executables can be shared by contacting the senior author.

- 4. Line 73. Give the reasons why the "3.07 g/cm³ iso-density surface" was selected for the Moho.**

We elaborate on our choice in the revised manuscript. We used 3.07 g/cm^3 as the iso-density surface, which has been shown to accurately fit the seismic Moho when performing minimum-structure inversion modeling^{19,21,26}. This contrasts with the commonly used surface-based Parker-Oldenburg modeling method^{27,28}, that solves for the depth of an interface between an assumed homogeneous-density crust and mantle. Due to the known heterogeneity in density in the Abitibi crust³³ we opted not to use the surface-based modeling method, instead using the more sophisticated, albeit more computationally demanding, minimum-structure inversion method.

- 5. Line 78. Explain what the “signal” is that was output from the inversion code. The measured Bouguer anomaly is referred to the geoid and assumes in its reduction a certain density for the rock between the physical surface and mean sea level. Was the “signal” similarly referred to sea level?**

We have modified the figure caption to explain the nature of the signal. The inversion code outputs the forward signal of the inversion model. The difference between the inversion model's forward signal and the measured Bouguer anomaly is the model residual, which is plotted in this figure. The forward signal of the inversion model would be similarly referred to sea-level.

- 6. Line 100/101. The Moho derived from the inversion model is compared to CRUST1.0, a teleseismic model and five Lithoprobe seismic lines. However, there are issues associated with each of these sources of seismic data. There is [1] no evidence provided in the paper that there is any active source data available within the region of the study area. The teleseismic data is limited to Receiver Function data which is based on an assumption of P wave crustal velocity between the physical surface and the Moho. There is [2] no evidence provided in the paper of what velocity was used or discussion on whether or not this velocity is an appropriate one to use in the study area. Finally, the Lithoprobe data are seismic reflection profiles and so are in two-way travel time, not depth. There is [3] no evidence provided in the paper of what velocities were used to convert the reflection data to depth.**

- 1) We provide evidence in Fig. 3 showing the locations of each of the seismic surveys (which were located within the Abitibi), and the citation to each of the respective studies.
- 2) Each of the published surveys used in the paper provides details on the processing and modelling that were done to resolve a Moho

interface. The seismic studies all focused on the Abitibi and used velocity models best tailored to this region's crust.

- 3) All the velocity information is provided in the original studies. The reflection profiles are originally in two-way travel time, but they were converted to depth as in other Lithoprobe studies.

7. **Figures 3 and 4. The relationship between the observed Bouguer anomalies and the inferred Moho undulations is very difficult to see in the paper. At the very least profiles of the observed Bouguer anomaly should be shown in both Figures 3 and 4 so that the reader can see the relationship.**

We have added the Bouguer data near the Figure 4 profile to better show this relationship. Since this is not a geometric inversion there won't be a 1:1 relationship between the Moho topography and the Bouguer anomaly. We show that many density variations in the crust contribute to the overall Bouguer anomaly signal (see the new Fig. S4). We hope the change to Fig. 4 and the new Fig. S4 clarify the relationship.

8. **Line 105. Was it not assumed in the model that the sub-surface density structure was layered?**

We did not assume that the sub-surface density distribution in the Abitibi was layered. As explained in the "Gravity Modelling of the Moho" section, our modelling produces a 3D density distribution in the sub-surface, from which a contour level can be isolated to represent the Moho. But the model is not itself a collection of layers. To help clarify how we parameterized our model we added a figure of the model mesh to the Methods section, showing the distribution of "pixels" that made up the model. We also added further description on the parameterization used to produce the model.

9. **Acknowledgments. Makes no mention of data availability or access to the inversion software used in the paper. Web links should be provided that link to the Bouguer anomaly and topography data and to the vidi inversion code.**

We have added these components to the Data Availability and Code Availability sections.

10. **In summary, the reviewer had many concerns about the validity of our results because of the lack of forward modelling.**

To address the reviewer's concerns, we performed forward modelling to show that the undulations in the Moho produce a signal much greater than

the model residual, and therefore is significant. However, we believe these concerns are related to a misunderstanding of the parameterization used in our modelling (see response to comment #2). Because we did not perform a geometric inversion, we were not solving for a (or a collection of) iso-density layer(s). If we had, indeed forward modelling the signal from each layer would inform how each layer contributes to the total signal. Instead, we solved for a 3D distribution of density throughout a model of tetrahedral cells. Isolating the component of the model that lies below the resolved Moho, and forward modelling its signal shows the undulations in the Moho (and density anomalies in the upper mantle). The anomalies have amplitudes of 20-30 mGal, well above the model residual, which has a mean of 0 mGal and standard deviation of 2 mGal (see the new Fig. S4).

Regarding the undulations observed in the Moho, the many seismic studies performed in the Abitibi have all shown changes in Moho depth, as we showed in Fig. 3B, which we verified our model against. Our model would be wrong if it presented a flat Moho. If forced to have a flat Moho, the model would still produce a density distribution in the crust to fit the observed Bouguer anomaly data. However, we verified our model depths to those previously published (and derived from methods independent of gravity modelling).

Manuscript NCOMMS-24-59268B - Archean rifts and triple-junctions revealed by gravity modelling of the southern Superior Craton.

Reviewer 1 comments

Introduction

The revision of this paper by Galley et al represents an improvement on the previous version. The majority of the revisions are excellent – however a few further comments are provided below. This paper deserves to be published in Nat Comms, but it needs to be more exact in its points in key places. Parts of the paper and its major points on tectonics are still far too vague.

This is a good paper, but it could be a great paper with a bit more work and care and attention. There are numerous typos and errors, and the prose itself could be clearer in places. However, the science is sound and I think this will be a really nice contribution after these revisions. I envisage that after these relatively straight-forward revisions the paper will be suitable for acceptance.

Response to previous comments

I do not particularly like the way the authors have summarised the reviewer's (my) points and then made their comment or rebuttal. Please paste the actual comment, verbatim, into your report, and then address it. This avoids any ambiguity; the editor and I can see the original comment next to the response.

Apologies. This has been changed.

In general, the revisions were thoughtfully performed and comments addressed or rebutted well. However, I have further questions regarding the responses below:

Thank you for adding figure 5. This is excellent and I think is a very powerful link between the geology, geochemistry, and geophysics. This is something that is, unfortunately, quite rare. I suspect this figure will be used a lot.

Thank you.

I still find the microplate argument ambiguous. Were these pieces of crust actually totally separate plates? Or different terranes? I am not sure... but I think calling them microplates will at least generate some interesting discussion. Your definition is progress, but the rifts surely have to be 'mantle tapping'? i.e., like at

plate margins? I wonder if the situation in East Africa is a good analogy. Some discussion on the size and geometry of microplates needs to be in the main text. Apologies if I missed it.

We have expanded the description of microplates in the introduction

“Bird (2003) defined modern microplates as small, mostly rigid areas of lithosphere that range in size from 1000s up to 1,000,000 km². Detailed studies of today's oceanic microplates show that they behave as independent plates but are continuously deforming structures (MacLeod et al., 2017; Baxter et al., 2020).”

You say “the abstract is re-written” but the track-changes show only four changes that are extremely cosmetic. The two statements you make in the abstract are not factually correct (lines 9-10 and lines 20-21) and these need to be revised or toned down. It will not affect your ms as far as I can see but be more accurate to current Archean geoscience. The reality is, we cannot be sure of many things in Archean geology and geodynamics, and hence we need to be open about that as we try to push things forward.

I think I made this point in the first review – the first line of the abstract is not appropriate (line 9-10). It makes a statement as fact, that we do not know. We do not know that plate tectonics started at 2.7 Ga – this is not a fact and needs to be removed or edited. I'm sorry, but I insist on this. This is because there are a whole range of opinions on when PT started, from the Hadean, Mesoarchean, Neoarchean, to Neoproterozoic. Even if you mean “non-subduction, mobile-lid plate tectonics”, this statement will not necessarily be read this way – if you do mean this, you should at least clarify it. Furthermore, this would also be debated – some think a stagnant lid is appropriate for the Neoarchean. I do not see why microplates and rifts could not exist in a stagnant lid, with localised extension taken up by ‘crumpling’ at the margins, similar to Phanerozoic ‘intraplate orogens’ such as the Alice Springs

Orogen. Again, this is what is theorised for Venus. A connection of rifts does not necessarily mean moving plates – if you want to infer this, you probably need further evidence. Referencing some paleomagnetism from Archean rocks would be a start, such as these papers:

<https://pmc.ncbi.nlm.nih.gov/articles/PMC7176424/>

https://www.sciencedirect.com/science/article/pii/S0012821X0700516X?casa_token=DtodWmKuBKkAAAAA:m3VPwWcldcMmDoZTCtttjIHUPMWgYhWKBOT2fOHDYOjj_2PxuoNDRNMycU4YqaBYfDIXx8lyDA

They show evidence of plate movement in the archean, although as you can imagine, this is contested. I personally like the idea of episodic subduction, and this matches the data in Mole et al 2021 for the SE Superior. However, these rifts appear to be ‘pre-subduction’, if subduction started at 2.695 Ga in the Abitibi, so in that context they would potentially need another driver.

On top of this stuff, some of the hottest magmas on Earth are known from the Neoproterozoic, which suggests this may have been a thermal peak, rather than a cooler inflexion (see Herzberg et al 2000). You can say this, but not as a statement. You need to add ‘may’, or ‘possibly’ etc. If you feel this is a result of your study, that’s fine, but don’t open with it. Right now, as I said in the previous review, this is not correct, and a poor way to start the paper.

You might rather say that “The nature of Archean global tectonics, and the associated geodynamic regime, is much debated in modern geoscience, despite decades of research. In this study, we use a new geophysical dataset to show that, by the Neoproterozoic, convective forces etc...”

That way you can state lines 11-14 more as your result and findings, which they are.

We have modified the first sentence of the abstract with something similar to what the reviewer has suggested, avoiding a categorical statement about the start of subduction at 2.7 Ga.

“The nature of Archean tectonics and the associated geodynamic regime is much debated in modern geoscience, despite decades of research. In this study, we present a new geophysical model to show that, by the Neoproterozoic, convective forces ~~Towards the end of the Archean, approximately 2.7 billion years ago, the Earth’s crust had sufficiently cooled to begin a transition to plate tectonics. Convective forces~~ from rising mantle plumes or early forms of plate subduction caused widespread extension, creating linear zones of crustal growth.”

Where appropriate, we have replaced "horizontal tectonics" with "mobile-lid tectonics" to avoid suggesting any inferences about the drivers.

“The connectivity of the interpreted relict rifts is possible evidence for the existence of Neoproterozoic plate boundaries and triple junctions and supports a model of ~~at least local horizontal~~ mobile-lid tectonics during this stage of Earth’s history.”

Apart from these corrections, we do not believe the full discussion of the topic suggested here by the reviewer can be reasonably covered in the abstract.

Line 18: timing is important for the Abitibi – is this model strictly for the syn-volcanic period i.e., 2750-2695 Ma? If so, a few lines on how the compressive phase 2695-2670 Ma influenced it might be useful to complete the circle. You say in the abstract your microplates are evidence for horizontal tectonics (i.e., compression) but there is no evidence for this structurally until ~<2690 Ma – how does this work?

We have addressed this further into the manuscript, in the last paragraph of the Introduction. There we discuss the compressive phase the Abitibi experienced, how that affected the upper crust and how, with the imaging we have at our disposal (seismic), the lower crust/Moho was not similarly affected. As noted above, we have replaced “horizontal tectonics” with “mobile-lid tectonics” to avoid any inferences about a particular geodynamic regime apart from synvolcanic rifting.

Lines 19-20: You have found rifts – or plate boundaries in a mobile-lid tectonic setting – this not direct support for horizontal tectonics. Archean tectonics is littered with these kinds of statements. Report on what you actually have – rifts are evidence for extension. Horizontal tectonics will be interpreted as subduction – if you don’t mean this, then state ‘mobile-lid’ or whatever you do mean.

Done. As noted above.

Lines 25-26: you have not referenced this. Please be more exact when describing what you mean by plate tectonics – just mobile lid or also subduction? – if the latter, say ‘modern-style plate tectonics’. Rifting does not = modern-style plate tectonics, so don’t infer that. If you mean mobile-lid and horizontal movement – then be specific. Don’t leave ambiguity. This a massively important point – be very clear on what you mean.

Done. As noted above.

“A proposed lack of ~~mobile-lidplate~~ tectonics was suggested, in part, from an apparent lack of Archean rifting (Hamilton, 1998).”

Line 35-36: I do not understand why you keep making statements. There are very few things in Archean geoscience that can be said with certainty, as a statement. This ‘may’ have been a pivotal time – we don’t know for certain – new data suggests the Neoproterozoic was pivotal. Don’t state facts that we don’t have for the Archean. Half of these statements would be corrected if you just inserted ‘may’ or

'likely'. It's a problem in Archean geoscience that two papers next to each other can make two contrasting absolutist statements that contradict each other. It's made the science messy, don't be one of them.

The wording has been changed to avoid this problem.

"The Meso- to Neoproterozoic is seen **by some** as a pivotal time in Earth's history, **that was somehow** transitional between stagnant-lid and mobile-lid tectonics (Brown et al., 2020; Palin et al., 2020)."

Ah ha! Line 43 we get to the point! So, when you say mobile-lid, you DO mean mobile-lid +/- subduction. You should remove references to 'horizontal tectonics', and just say mobile-lid. Horizontal tectonics is disliked as a term by many (JF Moyer for one) and it is mostly associated with modern-style plate tectonics. You can remove all ambiguity by simply stating 'mobile-lid' tectonics. Many see horizontal vs vertical as problematic language, when it is likely to be a dynamic continuum.

Done. We have removed "horizontal tectonics" and replaced it with "mobile-lid tectonics" throughout the manuscript.

Of course, relatively small-scale rifts of this nature do not necessarily mean there was a global mobile-lid either... but at least local mobility. On Venus rifts are taken up by localised folded mountains, a system known as "tesserae" – these are regional dynamic forces in the mantle, and not associated with a true mobile-lid. In fact, you should probably mention such features as part of your interpretation – they are a valid interpretation for what we see in the Archean. I think it would be really valuable to clearly present all these options, even as a list if it's easier, and then choose your preferred option (with reasoning). You have done this in the 'origin of rifting' section, but you could do more. I suspect you have more words available – don't be afraid to add more detail.

A full discussion of this topic is far beyond the scope of this paper. We offer one piece of evidence for Archean rifting. We also emphasize that our conclusions do not necessarily support a global tectonic style, but a more local phenomenon for the Abitibi. We have emphasized this the end of the introduction:

"Although rifting by itself does not prove a large-scale **horizontal mobile-lid** tectonic regime, the existence of crustal fragments that are encircled by rifts are a strong indication of **at least** a local microplate **regimemosaic** in the Abitibi."

Line 50: parts of the Superior are younger than 2690 Ma.

We are referring only to the syn-volcanic stage of the Abitibi was between 2.75 and 2.69 Ga:

“The Abitibi greenstone belt, situated in the southern Superior Province of Canada, is the largest and one of the best preserved Neoproterozoic greenstone belts in the world (Monecke et al., 2017) (Fig. 1). It formed between 2.75 and 2.69 Ga in a series of submarine volcanic events thought to have been centered in Archean rift basins.”

Line 58 – spelling error ‘ruft’

Apologies. Corrections were made in the final version of the manuscript but not in the "track-change" version.

Line 64 – change to ‘mobile-lid’

Done.

Line 85 – Mole et al 2022 should be here, Mole et al 2021 should be at the end of this line. Mole et al 2021 is the original source of the isotopic and WR dataset. Use Mole et al 2022 for any mineral systems inferences. Some mistakes across the ms with these references.

Mole et al. (2022) was the reference used here. This has been corrected throughout the rest of the manuscript.

Line 119-124: It is good you have made this point, but it feels like a ‘we hope this isn’t the case’ – can you be more specific? It seems a key point – can you prove that we aren’t just seeing the surface effects? Apologies if this is already in the Methods. Maybe a reference here would help.

We have added the reference to the new Hayward et al. (2024) study that produced a gravity inversion model of the Pilbara. There they show, using the same type of modelling, that near-surface mafic volcanic belts, felsic plutons, then density features in the mantle can all be modelled independently of each other.

Line 131: if the resolution of your model is 10 km, and the range in moho depth is 32-45 km (line 149) – this range is very close to the uncertainty... does this affect how resolvable the features are?

Thank you. I should have specified 10 km “horizontal” resolution, not vertical, which was the largest spacing between the gravity stations that I modelled. The vertical resolution is controlled by the cell size of the model as shown in Figure S1 and described in the supplementary data. They were generally 3 km wide (the tetrahedra aren’t regular, so there’s minor variation in their dimensions) and 3km tall. Unlike cubic cells that are stacked on top of each other, tetrahedra can vertically overlap, so the vertical resolution of the model is much less than 3 km and closer to 1 or 2 km.

Line 153: partial melting of the crust

This is corrected.

Line 161: good point

Thank you for making that suggestion in the last round of reviews.

Line 175: Matheson

Thank you. Another error not changed in the tracked document.

Line 180-181: length is used twice here, re-write

One of those should have been a width:

“(between 30 and 50 km and in ~~length~~ width between 60 and 130 km in length)”

Line 183: time? (spelling)

Thank you. Another error not changed in the tracked document.

Line 183: some actual examples here would be useful, rather than just citations.

We have added some:

”Although regional north-south shortening has likely affected the overall geometry, the lengths and widths of the inferred rift zones (between 30 and 50 km and in length width between 60 and 130 km in length) are comparable to modern continental (e.g., East African Rift: Rosendahl, 1987) and oceanic rifts (e.g., in the Lau Basin: Galley et al., 2024) (Galley et al., 2024; Rosendahl, 1987).”

Line 185: hmm is this true? A stronger argument might be that the Archean crust was hotter, and therefore behaved differently during compression – see the paper below:

<https://www.nature.com/articles/s41561-018-0138-0>

I suspect that the rift zones were more likely to survive in a crust that collapsed laterally, rather than thickened vertically. This might provide some better evidence for your point.

Line 230: apologies, you have already seen the Calvert paper. I still think you might look to use it in the point I mention above. Apologies if I already said this stuff last time!

We have clarified this statement, adding that lower crustal flow does not change the topography of the Moho.

“At the same time, we expect the **relative** topography of the Moho to be largely preserved through the deformation owing to the ductile behavior of the lower crust. **The lower Archean crust has been shown to flow under stress (Calvert and Doublier, 2018), a condition that leads to the thickening of the lower crust when undergoing post-rifting compression. However, the shallowing of the Moho is still preserved despite the relative post-rifting thickening of the lower crust with respect to the upper crust (Bertotti et al., 2000).**”

I’m not sure if you mention it, but it seems intuitive that the width of your mapped rifts are minimums, and that they have been closed at least partially by the later compression.

Yes, we state “Although regional north-south shortening has likely affected the overall geometry, ...”

Figure 5: As I said – this is a really nice figure. Combining the separate systems endorses your model and validates those existing models into a holistic system.

However, for some reason you have cited Mole et al's Yilgarn paper for these data. The correct reference should be the Mole et al 2021 paper (Precam Res). Mole et al references seem to be mixed up a bit in their usage: The isotopic/geochemical dataset itself is from Mole et al 2021. When using it to refer to mineral systems, use Mole et al 2022. If you want to compare your dataset to similar systems in other cratons, or cite an example of other Archean rifts, use Mole et al 2019 (i.e., Kalgoorlie-Kurnalpi Rift; KKR).

Thank you for that correction.

Line 214: be careful when using 'most reasonable' – I know what you mean, but unfortunately what is reasonable is subjective... Better to say, something like, 'in conjunction, the geophysical and geochemical evidence best supports a rift model....'

We have changed this first sentence of the Discussion to:

"The presented geophysical, geologic, and geochemical evidence best supports an extensional model of the Abitibi where ~~The most reasonable interpretation of~~ the linear zones of shallow Moho ~~in the Abitibi is that they~~ are the root zones of relict Archean rifts."

Line 279: you need a reference here

We have added the citation:

Jackson, S. L., & Cruden, A. R. (1995). Formation of the Abitibi greenstone belt by arc-trench migration. *Geology*, 23(5), 471–474. [https://doi.org/10.1130/0091-7613\(1995\)023<0471:FOTAGB>2.3.CO;2](https://doi.org/10.1130/0091-7613(1995)023<0471:FOTAGB>2.3.CO;2)

Line 295: "Ey" should be Eu

Corrected, thanks.

Lines 292 to 295: you have these Eu/Eu*/Y relationships backwards. Values >1 indicate a deeper source/thicker crust (possibly – some contention on this) and <1 shallower crust. This is because, for deep crust we expect high Eu values (no plagioclase), and low Y values (garnet present). The high Eu = higher Eu/Eu*, and lower Y values result in a higher final number. Hence deep crust = >1. Vice versa

for shallow crust with plagioclase present, which takes out the Eu, no garnet = higher Y. See below figure from Mole et al 2022. You seem to have this correct in Fig 5D, so just an error in the text I think. Note the Ui/Yb map in Mole et al 2022 correlates quite nicely with your N and S rifts.

Corrected, thanks. That did appear to be just a typo in the text, the figure and its caption/legend was correct.

Line 291-298: This section is good but feels rushed. You talk about La/Sm, but in the Figure 5D, you have Sr/Y labelled (which is generally a depth indicator). Please make the figure and text consistent.

The low La/Sm could also indicate less crustal contamination i.e., from passing through thinner or no silicic crust. Contamination indicator (Th/Yb) correlates with decreasing eNd in Mole et al 2021, possibly suggesting the trend is due to contamination with an older, existing crust.

We have added the reference to Sr/Y ratios. The reference La/Sm ratios was intentional, but was not accompanied by a figure in this manuscript. We still wanted to mention the spatial relationship it has with our modelled rifts.

“In contrast, the interpreted rift zones lie generally below crust with a more depleted mantle source indicated by ϵ_{Hf} values (Fig. 5C), thinner crust (i.e., $(\text{Eu}/\text{Eu}^*)/\text{Y}^*10,000$ ratios $\ll 1$; Fig. 5D) (Mole et al., 2021), **more shallow melts (low Sr/Y ratios; Fig. 5D)**, and less fractionated melts (low La/Sm ratios (Mole et al., 2021)), all consistent with rifting of the Archean crust (Mole et al., 2021, 2022).”

Line 307 – again clarify what you mean when referencing horizontal tectonics. This last section is where you need to be very clear what you are inferring; is it mobile-lid tectonics? Papers by Gerya et al have shown how plumes can form mobile lids, which then induce short subduction events – this might fit the timeline of events in the Abitibi:

As suggested in previous comments we have changed “horizontal tectonics” to “mobile-lid tectonics” to avoid this confusion.

Origin of rifting section: I know what you are trying to say here but I think it could be more clearly written. I think removing vertical vs horizontal tectonics will help it be more specific – talk about the processes you mean directly i.e., sagduction/drip tectonics and the other options discussed in this response. If

you are talking around mobile microplates, then say that. You have done a good job and trying to extricate the data from going too far, which is great, but I think it needs a bit more polish. For Nat comms, you need to provide more, just saying it means horizontal tectonics is too vague – show us how well you understand the complexities of Archean geoscience and go further with your model/process.

Remember, just because you have evidence of regional mobile microplates does not mean it was a completely mobile-lid – as I said before, Venus has evidence for this and is generally seen as a stagnant lid. I think ‘locally mobile-lid’ covers pretty much all bases. In the plume model above (Gerya et al), the rifting is actually driven by a vertical process, with lateral spread of the ‘plates’ – so I don’t think committing to horizontal tectonics is necessary or even correct at this point. I think you could include this, or at least present this further alternative.

If you must use “horizontal tectonics” at least use it after the explanation you provide. That would mean removing it from the abstract to avoid confusion.

Done. As suggested in previous comments we have changed “horizontal tectonics” to “mobile-lid tectonics” to avoid any assumptions about the driver of extension. In the following statement, we clarify our understanding of the significance of the geophysical modelling relative to different notions of horizontal or vertical tectonics already discussed extensively in the literature.

“Rifting in a modern plate tectonic context is often linked to extensional forces driven by plate subduction, whereas in the Meso- to early Neoproterozoic other processes may have been more important, ranging from sagduction/drip tectonics (e.g., Mareschal and West, 1980; Collins et al., 1998) to plume-driven processes (e.g., ~~it is believed mantle plumes and mantle convection drove extension~~ (Gerya et al., 2015; Harris and Bedard, 2014; Piccolo et al., 2020). ~~Rifting could also have occurred in regimes exhibiting only sagduction/drip tectonics driven by density contrasts in the crust (Mareschal and West, 1980). The emplacement of dense mafic material over less dense crust and partially melted lower crust leads to the sinking, or sagduction, of the mafic material (Collins et al., 1998) (so-called dome and keel structures). Rigid crust, typical of horizontal~~ The results presented here provide evidence of at least local mobile-lid tectonics ~~Mobile-lid tectonic regimes allows for~~ where rifting was ~~to be~~ focused along discrete corridors, whereas ~~more ductile crust in vertical diapirism and sagduction driven~~ other tectonic regimes leads to disseminated extension across large areas (Choukroune et al., 1995). The Moho topography of the Abitibi best fits a ~~horizontal~~ mobile-lid tectonic regime with distinct corridors of shallow Moho encircling regions of thicker crust (Fig. 5). ~~This contrasts with the coupled shortening and extension of the crust associated with diapirism and sagduction in a vertical tectonic regime (Johnson et al., 2014).”~~

Line 320-321 – you use ‘although’ and then ‘however’ in next sentence – these are synonyms, reword this.

This sentence was replaced in the conclusion.

Line 321: why? What about the model suggests this?

This question is addressed in the concluding remarks.

Line 323: don’t need ‘widespread’ here, it’s ambiguous and I’m not sure you have evidence for it.

It has been removed.

**Line 323: This is not the best way to end the paper. If you have words left, a paragraph bringing the reader back to the main point, and its implications, would leave a better impression. Perhaps use the last sentence to summarise the discussion of different models/ideas from the section above? Or leave the reader with what further implications may be? i.e., “ our results show a strong empirical relationship between mantle plume activity, microplate formation, and major mineral systems for the Abitibi, which imply at least a locally mobile-lid tectonic environment existed in the Neoproterozoic. These results have implications for the formation and evolution of all cratons, as well as their host mineral systems.”
Something like that?**

We have revised the last paragraph to highlight the possible implications of our findings for the larger audience.

"This study presents a new gravity-derived model of the Moho surface beneath the Abitibi greenstone belt that reveals a possible Neoproterozoic rift network. The geometry of the proposed microplate mosaic is supported by trace element and isotopic data of associated volcanic rocks and the occurrence of rift-related mineral deposits. The results show a strong empirical relationship between mantle plume activity, microplate formation, and major mineral systems in a local mobile-lid tectonic environment."

I think it would also be worth making the point that deep sea hydrothermal systems have been associated with these rifts, as exemplified by the VMS

systems. The origin of early life may be connected to these type of settings, and hence these new maps of Archean rift systems map out potential locations for early life i.e.,

<https://www.sciencedirect.com/science/article/pii/S0301926814001454>

This might be a nice way to end the paper, to show it has profound implications for the wider Earth system and environment.

There is also a new paper in Geology (screen shot of a figure below):

<https://pubs.geoscienceworld.org/gsa/geology/article/53/3/269/650953/Inception-of-ridge-ridge-ridge-triple-junctions>

It might be good to reference and/or include this evidence for rifts in an arc environment, in the interest of balance. It doesn't mean you need to commit either way, but it would mean you present possibilities for multiple settings. It also supports your 'mobile-lid' / 'laterally-dynamic' tectonic process. Perhaps with the komatiites occurring at the same timing of rifting, and komatiites connected to plumes (ambient mantle was too cool; Nisbet et al.), the plume model is more likely; but you should decide what you prefer.

Thank you for these suggestions. We added the particular references and suggest possible implications of a mobile-lid regime within the Abitibi in the previous revised paragraph. However, we prefer to focus on the merits of our modelling for others to consider rather than a review of the literature on Archean geodynamics. We trust that knowledgeable readers will grasp the significance for early Earth tectonics, mineral systems, and "origins of life" without a full discussion of all the models here.

Major changes required:

Remove all occurrences of 'horizontal tectonics' – replace with what you mean, which is 'local mobile-lid tectonics'. I know what you mean, but horizontal tectonics as a term is closely associated with subduction and modern-style plate tectonics, so better to use something else.

Done.

There are other possibilities you need to explain and explore, to make the origin of rifting section more thorough and less vague.

We have expanded on the first paragraph in this section with reference to rifting in a modern context and (in the revised paragraph) with reference to other processes ranging from sagduction/drip tectonics (e.g., Mareschal and West, 1980; Collins et al.,

1998) to plume-driven processes (e.g., Gerya et al., 2015; Harris & and Bedard, 2014; Piccolo et al., 2020).

As stated in the first review, do not make statements that are not currently valid in Archean geoscience. The reality is we don't know enough on many aspects to make statements – for most points there is a counterpoint. This is easily fixed by qualifying statements by adding 'may' or 'likely' to take the absolutism out of the sentence.

Thank you for helping with this and in refining the manuscript and its message.

Tidy up the quality of the writing – it feels a bit rushed. There are numerous spelling mistakes, and some sentences could be clearer. Be very careful when presenting geochemical data – there are some major blunders in there.

We have made an effort to polish the manuscript. While there regrettably were some spelling mistakes in the “Tracked” version of the document, they were all caught in the final un-marked version.

Reviewer 2 comments

The authors made appropriate revisions to the paper, which is now suitable for publication.

Taras Gerya, Zurich, 17.02.2025

Thank you.

Reviewer 3 comments

I have looked over the revised paper of Galley et al. Unfortunately only a few of my comments have been addressed.

The assumption of 3070 kg/m³ for a mantle density has still not been justified. The density, which is more typical of lower crustal is too low for the mantle – it is even lower than the density (3180 kg/m³) normally assumed for the mantle at an actively spreading mid-ocean ridge!

The 3070 kg/m³ density contour is not assumed to be the density of the mantle, but rather the density at the interface between the lower crust and the mantle. You are correct, typically when the upper mantle is assumed to have a homogeneous density it is closer to ~3300 kg/m³, and a homogeneous crust can be typically ~2670 kg/m³ for the upper crust (continental) and ~2900 kg/m³ for the lower crust. The density at the interface between the lower crust and the mantle would be represented by a value somewhere between that of the upper mantle and the lower crust; i.e., between the 2900 kg/m³ lower crust and 3300 kg/m³ upper mantle, or 3100 kg/m³. Model testing done by myself and others (such as Dr. Kim Welford who advanced the gravity modelling method for a heterogeneous density model of the crust/upper mantle to resolve the Moho) have found that the 3070 kg/m³ contour most accurately fits the Moho.

“The Moho interface was then extracted from the 3D density inversion by isolating the 3.07 g/cm³ iso-density surface. This iso-density layer has been shown to be an accurate proxy for the Moho when performing minimum-structure inversion modeling (Galley et al., 2024; Welford, 2024; Welford & Hall, 2007), in contrast to the commonly used surface-based Parker-Oldenburg modeling method (Oldenburg, 1974; Parker, 1973). The latter solves for the depth of an interface between an assumed homogeneous-density crust (e.g., 2.85 g/cm³ (Chappell & Kuszniir, 2008)) and the mantle (e.g., 3.33 g/cm³ (Chappell & Kuszniir, 2008)). We opted not to use the surface-based modeling method, instead using the more sophisticated, albeit more computationally demanding, minimum-structure inversion method (see Methods for details), to account for the known heterogeneity in the density of the Abitibi crust (Esmail Eshaghi et al., 2023). **Our more heterogeneous density model would then resolve a Moho interface at a density value between that of the crust and mantle.**”

The inverted Moho is still being compared to travel-time seismic reflection data. There should be an explanation of how these data have been converted to depth and what the uncertainties on these depths might be. Readers should not have to search through the literature for this information.

We have added more information on the processing of the seismic reflection data to the section “Comparison with Seismic Profiles”, as well as more details for the other two models we used to compare with our gravity model. We also added a comparison to a seismic refraction model of the Moho to the same section and Figure 3.

“The results of the gravity inversion are compared to Moho depths from the Lithoprobe reflection seismic experiments (John Ludden & Hynes, 2000), a Lithoprobe seismic refraction model (Grandjean et al., 1995), those from the CRUST 1.0 model (Laske et al., 2013), and from teleseismic data (Rondenay et al., 2000) (Fig. 3B). The reflection seismic sections were depth-migrated using a constant velocity model of 6 km/s, creating an approximate depth model of the Moho and overlying crustal features (Ludden & Hynes, 2000). The seismic refraction line modeled by Grandjean et al. (1995) resolved a Moho line from a 2D velocity model of the crust and upper mantle, providing more accurate depth information on the Moho. The teleseismic measurements led to a series of velocity models of the Abitibi crust and upper mantle, (Rondenay et al., 2000). The CRUST 1.0 model is the coarsest of the Moho models with a spatial resolution of 1 arc-degree (~111 km) and Moho depths derived from a velocity model of generalized Archean crust (upper crust: 6.2 km/s; middle crust: 6.4 km/s; and lower crust: 6.8 km/s) (Laske et al., 2013).”

The authors have now provided a Bouguer anomaly profile, but the vertical scale range of 1000-1060 mGal in Figure 4D makes no sense. I could not look up what the Bouguer anomalies are in the study area because none of the maps in the paper have latitude and longitude (they should do!) but Government maps show Bouguer anomalies in the range -20 to -100 mGal southeast of Hudson Bay.

We used “processed Bouguer anomaly data” and have noted this in the figures and text. In the section “Further Bouguer Data Processing” in the Methods we have explained what steps were taken to process the Bouguer data.

“To model the Moho topography in the Abitibi, we inverted processed Bouguer gravity data compiled by Natural Resources Canada (Canadian Geodetic Survey, 2022) (Fig. 2; see Methods for details on data processing) to produce a three-dimensional (3D) density model of the Earth’s crust (Fig. 3).”

The output of the inversion modeling is a density structure that has still not been verified by forward modeling. It should be a relatively straight-forward process to take the density contours such as those shown in Figures 3B and 4D and calculate the gravity effect of each contour using a line-integral method and the density contrast between layer above the contour and the layer below.

In summary, I am not convinced that the undulations in the Moho proposed in the paper are required to fit the observed gravity data. The authors need to demonstrate more clearly that the sources of the Bouguer gravity anomalies are not within the crust and that Moho is not flat.

In response to the previous review we did verify the inversion model with forward modelling, see Figure S4, and explained that the gravity anomalies resolved are much larger than the model residual, and are therefore significant and not just noise (the amplitudes of the forward signal of the mantle component range from ~20 to 50 mGal).

We did not produce a forward model of contours because that would be an approximation of our model. Our model is not composed of homogeneous layers separated by contours, but rather tetrahedral cells. So, it was more accurate to calculate the forward model of the cells that lie above the Moho (Fig. S4C) that represents the gravity anomaly produced from the crust, and compare that to the forward signal from the cells below the Moho (Fig. S4D). As suggested, most of the signal of the modelled data is fitted by features in the crust, but there is still a large, regional component that needed to be fit by deep features, which we interpret as the Moho/upper mantle.

With further explanation of the forward modelling (see below) we believe we have shown that the undulations in the Moho are required to fit the observed gravity data, that not all of the sources of the Bouguer gravity anomalies are within the crust, and that Moho is not flat. Density features in the crust were not enough to fit the modelled data, and that the interpreted undulations in the Moho are supported by the teleseismic, seismic refraction, and seismic reflection profiles.

We added a new section to the Methods section to explain this further:

“Forward Model Verification

To verify that the density anomalies in the upper mantle shaped the topography of the modeled Moho, we created a series of forward models of components of our inversion model (Fig. S4). The first forward model calculated the gravity signal due to anomalies above the Moho, i.e., crustal anomalies (Fig S4C). Next, a forward model was calculated from the component of the inversion model below the Moho, i.e., the upper mantle/Moho anomalies (Fig. S4D). Comparing the forward signals of the crustal anomalies and Moho anomalies to the modeled data (Fig. S4A) it is possible to quantify how much of the modeled data was fit due to density features in the crust versus features in the upper mantle. As expected, most of the data was fit by crustal density anomalies, but a significant, more regional component of the data had to be fit by deep, upper mantle density features. The amplitude of the forward signal of the mantle component was ~20 to 50 mGal (Fig. S4D), much greater than either the Model Data

Residual (Fig. S4B) or the uncertainty that averaged 0.3 mGal (Canadian Geodetic Survey, 2022).”

The source code of the computer program used in the inversion is still not being made publicly available. Readers will therefore be unable to understand how the code works or, more significantly, be able to reproduce the results in the paper.

The computer program is publicly available, but with the legal condition that it must be used only for academic purposes. As such, we cannot keep it on a repository for all to access at any time, but are more than happy to make it available to researchers by contacting Dr. Lelièvre, to verify this work or conduct different research. The code is currently in use at Memorial University of Newfoundland, University of Ottawa, Mount Allison University, and Woods Hole Oceanographic Institute.

From Nature’s reporting standards and availability of data, materials, code and protocols:

“An inherent principle of publication is that others should be able to replicate and build upon the authors' published claims. A condition of publication in a Nature Portfolio journal is that authors are required to make materials, data, code, and associated protocols promptly available to readers without undue qualifications.”

We feel like we have met these requirements.

Reviewer #1 (Remarks to the Author):

This is the third time I have reviewed this paper, and I am now happy for the paper to be accepted. Thank you to the authors for making the corrections in good faith. I think this will be an interesting contribution. However, as I said in my initial review, this is based on the assumption that a geophysicist has reviewed and cleared this manuscript, as I do not have the expertise to check that element of the paper.

A few final comments:

Line 30-31: remove somehow, add 'potentially'

Done.

Line 162: This should be 'deeper' level of erosion - if these areas represent the 'horsts' then they eroded more deeply to reveal the intrusive rocks (as they where at a higher level), whereas the grabens or basins still contain the volcanic sequences. Line 163 can still be true in this scenario. However, what you say in line 164 suggests you already know this, so perhaps just changing the 'levels of erosion' to 'levels of subsidence' is more accurate.

This should have been “deeper”, thank you.

Line 297-299: Nice way to finish!

Thank you!

Reviewer #3 (Remarks to the Author):

I have looked over the revised paper of Galley et al. Unfortunately only a few of my earlier comments have been addressed.

1. The 3070 kg/m³ density contour. I checked the Galley et al. (2024), Welford (2024 and Welford & Hall (2007) papers but could not find any reference in them to the 3070 kg/m³ contour as Moho. The Parker methodology is the most commonly used way to compute the effect of Moho because seismic refraction data indicate that the transition from crust and mantle is generally associated with abrupt change in density, not a gradual one. At the very least, a comparison should be made between the model predictions and one assuming the crust and mantle had homogeneous densities of, say 2850 and 3300 kg/m³ respectively.

To address the reviewer's concerns about the 3070 kg/m³ density contour, we have added to the manuscript: "Previous studies similarly modeling the Moho have used the 3020 kg/m³ iso-density contour^{25,26,32}, however through further testing we found that the 3070 kg/m³ contour better fit known Moho depth in the Abitibi." Lines 90-92.

We agree that the Moho in Archean terranes (as observed at present) is more abrupt than the Moho in the modern oceanic crust, for example. To address this, we previously added a second modelling step outlined in the Method's section "A Priori Moho Structure Constraints". Our final density model does, in fact, represent the Moho as a discrete change in density. This was not clear in earlier drafts, in part due to a lack of explanation in the main text (it was only in the Methods section) but also because of the choice of colour bars emphasizing density variation in the crust but washing out the Moho and mantle. Figures 3 and S2 have been adjusted to better explain the sharp density boundary across the Moho. We have also provided the additional explanation in the main text, under the "Gravity modelling of the Moho" section (see lines 92-115).

We chose not to construct another density model composed of a discrete Moho interface separating a homogeneous crust and mantle (with respective densities 2850 and 3300 kg/m³). It is known that there are large variations in density in the crust (reflecting the greenstone belts, felsic plutons, and ultramafic intrusives), and so it would be incorrect to assume a homogeneous density. Such a model would incorrectly require density variations in the mantle topography that likely don't exist.

2. Comparison with seismic data. The absence of controlled source seismic refraction data remains an important missing constraint on the gravity modeling. I am not convinced that seismic reflection profile data can provide any useful constraints on Moho depth in the absence of detailed velocity information. Teleseismic data can provide estimates of Moho depth, albeit for an assumed average velocity of the crust, but requires closely spaced seismic recording stations and techniques such as H-k stacking, which I do not know were

employed in this case. Finally, CRUST 1.0 is a 111 × 111 km grid of existing, mainly seismic refraction, data which be of limited use in resolving the subtle, short wavelength undulating variations in Moho depth inferred in this paper.

We still do not understand the reviewer's comment that seismic data are lacking. The refraction profile was included in Figure 3 with the edits from the last review, and discussed in the "Comparison with seismic profiles" section. We also further emphasized that the CRUST1.0 model is regional in resolution, and discussed how the assumption of a homogeneous crust with respect to velocity can lead to misleading Moho features in the seismic profile. To emphasize this, Figure 3 was updated to include a second profile better aligned with the refraction and teleseismic line. The colour bar was also fixed as to not wash out the density in the mantle, but rather show variations in density across the whole model. Previous versions of the colour bar obscured the sharp transition in density across the Moho, which was causing confusion.

For added text see section "Comparing with Seismic Profiles", lines 125-153.

3. Observed and 'processed' Bouguer anomalies. I am afraid that simply adding the word 'processed' to 'Bouguer anomalies' that are of the order of 1000 mGal makes no sense to this reviewer. The observed Bouguer gravity anomalies in the study region are approximately -20 to -100 mGal (according to Canadian government maps) and all figures that purport to show the observed Bouguer anomaly should reflect this.

We have revised all the data figures to show the original Bouguer anomaly. Although this was the data that was modelled, showing it the way we did in the manuscript has created too much confusion. Reviewer #4 made a similar comment. The use of the shift in gravity is further explained in our response to point 6 of reviewer #4.

4. Forward modeling. The modeling in Figure S4 is interesting, but it is not a verification of the density structure deduced from inversion by forward modeling. Rather, it shows the modeled contributions to the Bouguer anomaly of sources in the crust compared that of the mantle. Interestingly the figure, together with Figure 4D, shows how the inversion places so much of the source of the short wavelength gravity anomalies in the upper crust. The lower crust appears to make little contribution. Such a density structure for the crust in a cratonic setting is surprising and should be verified, for example, by spectral modeling. I was more confused, however, about what Figure S4 reveals about the long wavelength contribution. Is it mainly from the Moho (ie the crust/mantle boundary), the upper mantle or both? If the latter, then would we not expect a 'trade-off' between the two sources such that the Moho could be flat and provide no contribution while sources in the mantle provide the main contribution?

We stress that the modelled contributions of the crust and mantle to the Bouguer anomaly are forward models. To emphasize this, we expand on the forward

modelling of the mantle component by separating the signal due to the Moho from the signal below the Moho (see lines 449-470 and Fig. S3). In particular, we show that the forward model of the Moho produces significant gravity anomalies that were needed to fit the Bouguer anomaly data, and that density variations in the upper mantle below the Moho had little contribution. The additional forward modelling of the Moho surface shows that the density variations in the upper mantle are much lower in amplitude than the gravity anomalies produced from the modelled Moho, thus the "trade-off" mentioned by the referee does not appear to be the case with our model. See the Method's section "Forward Model Verification" for changes to the text and Figure S3 (also included in previous versions of the manuscript as Fig. S4).

The observation that the lower crust shows little contribution to the Bouguer anomaly (i.e., appears strongly layered with little lateral variation in density), has been addressed in the "Crustal Density Structure" section of the manuscript, lines 174-177, in particular with reference to Ludden & Hynes (2000), who also identified how the lower crust in the Abitibi appeared to be much more strongly layered than the upper crust.

5. Software availability. I repeat my earlier statement that the source code for the inversion method should be made freely available to all researchers so that the community has the opportunity to verify the methodology and repeat the calculations made in this paper.

It is freely available, so long as the researchers agree to only use it only for academic purposes. Details for making a request are provided.

In summary, my concern about the gravity modeling remains. Therefore, I question the main result of the paper: namely that the regions of shallow Moho reflect narrow linear belts of thin rifted crust that are bounded by the broader regions of thicker crust of one or more Archean 'microplates'.

We hope these latest revisions have adequately addressed the remaining questions of this reviewer. The main concerns about the gravity modelling were:

- the sharpness of the density change at the crust-mantle boundary
- the lack of controlled-source seismic refraction data
- the processing of the Bouguer anomalies
- the lack of forward modelling

We have addressed each of these concerns by:

- further clarifying that the Moho in our model is represented by a discrete change in density, through changes to the text and figures
- updated Figure 3 to better show that we had already provided controlled-source seismic refraction data
- revised all the data figures to show the original Bouguer anomaly and to avoid any confusion about the processing

- performed additional forward modelling to verify the gravity inversions

Reviewer #4 (Remarks to the Author):

Please note, that is not a complete review of the manuscript, but some thoughts on the gravity modelling performed as part of the study and the concerns by one reviewer.

As such, the modelling seems to be correct, but in my view is explained in unnecessary complicated matter.

1) Processed Bouguer anomaly

The processed Bouguer anomaly is simply a shifted Bouguer anomaly by adding a constant. Hence, I would recommend to show the Bouguer anomaly instead of the processed one.

Done. We originally thought showing the data we modelled was the more honest approach, but we recognize how that created confusion.

2) The shift has implications for modelling. Figure S3 illustrates that by showing that both models have the same gravity anomaly disregarding the shift (and I would omit this figure). Such a shift is often treated manually, by an average value or a reference model depending on the application. Here, a reference model would be most appropriate and this is sort of expressed by a 65 km thick slab of 400 kg/m^3 (and I suggest that authors use SI units for density). This shift in the inversion is somewhat similar to setting a reference depth in methods like Parker-Oldenburg.

We have removed Figure S3. The reviewer's description of the use of our "slab forward signal" is accurate. All densities in the manuscript have been reformatted to SI units.

3) This is also seen in the resulting density maps in Figure S2 (in which depth I wonder), which for result A shows that simple depth weighting is not sufficient. Only when the anomalies are shifted to positive values the inversion provides increasing densities with depth.

This is also correct. We have made changes to the text to better explain this shift as the addition of a reference model, rather than simply the signal of an infinite slab (see Method's section "Reference Model" and lines 113-115).

4) Coming to densities. A contrast of 400 kg/m^3 does not indicate a mantle density of 3070 kg/m^3 as it is well known that 2670 kg/m^3 is an okay value for near-surface density, but that density and velocity increase with depth (e.g. Zoback & Mooney 2002) and that there is a jump at the Moho. I have the feeling and this can't be seen from your colour scale, that you smear out the contrast (as densities in the mantle continue to increase). That would be fine, if you

demonstrate that the model response is the same, when all densities >3.07 are put to a constant value of 0.2-0.4 larger than the lower crust.

We have adjusted Figure 3 as to better show the Moho interface. We respond below to the concern about the use of the “infinite slab”, in how it can be better represented as a reference model (after comment #6).

5) “We opted not to use the surface-based modelling method, instead using the more sophisticated, albeit more computationally demanding, minimum-structure inversion method (see Methods for details), to account for the known heterogeneity in the density of the Abitibi crust”

This sentence is a bit a summary of the authors opinion, but I would argue that the word sophisticated is stretching it a bit. Yes, inverting for the density change with depth is okay and its advantage is that it provides crustal densities which appear reasonable. However, I would argue that you get the same result, when you A) filter the data, B) invert the long-wavelength part for the Moho (with reference depth of 35 km and density contrast of 400 kg/m³), and C) invert the short-wavelength residual for crustal internal variations.

We acknowledge that other equally sophisticated approaches could be used and have modified the text to reflect this (see lines 92-99). We chose the methodology described in this paper to avoid any filtering, which requires assumptions about what components of the data correspond to which features in the crust/upper mantle. We believe it is a strength of our 3D density model to be able to model the geometry of the Moho while simultaneously considering density variations in both the crust above it and upper mantle below it.

6) Furthermore, I disagree with Lines 381-388 and that you corrected for something done wrong in a normal correction. I applaud your honesty, that you state this correction as most modellers simply apply a shift, but this is also not a meaningless quantity, and we are simply speaking about a reference model, which also has isostatic implications.

We have revised our use of the word “correction” to avoid any confusion that we believe the classic gravity data corrections are wrong. Simply, we applied a “shift”, exactly as the reviewer describes it, in order to resolve a realistic density model of the subsurface.

We have changed the previous “Further Bouguer Data Processing” to “Reference Model” in the Method section to better explain the implications of adding the 1092 mGal signal to the Bouguer anomaly data in order to resolve a Moho interface that closely matches previously published seismic sections.

7) The inversion itself is well described, but the setting of the slab is actually the main factor for your results. I had to careful think about what actually has been

done and can completely understand the concerns and confusion of one of the reviewers.

We hope that we have addressed the concerns and confusion of reviewer #3, as described above and in the summary below.

In summary, I would omit Figure S3, put some of the technical descriptions on the inversion in the supplement, but discuss the role of the slab or reference model in the main text as this is the most crucial factor for the resulting models and refer to a normal use of anomalies. The processed Bouguer anomaly creates an unnecessary level of confusion and does not contain any new information. For gravity experts, shifts and reference models are nothing new. For non-experts that simply mystifies the gravity modelling.

We have removed Figure S3, and included more information on the role of adding in the reference model (previously described as a slab, see “Reference Model” section in the Methods). Adding the forward signal from the “65 km thick 400 kg/m³ slab” is now described in terms of a linear reference model. We have also omitted any mention of the processing of the “Bouguer anomaly” in the figures, as we now see how this could create confusion, as explained by the reviewer.

Review of “Archean rifts and triple-junctions revealed from gravity modelling of the Superior Craton” by Galley et al

Introduction

The paper by Galley et al presents interesting new data and perspectives on the crustal architecture of the Superior Craton. Understanding the relict architecture of these terranes can offer important snapshots into Archean crustal and tectonic activity. The paper is very clearly written and easy to follow. I think the dataset and the imagery provided are convincing and match other datasets that have imaged the Abitibi crust. However, the link to the plate tectonic debate does seem tenuous and I wonder whether it is strictly needed, given the impact of imaging relict Archean rifts alone.

I caveat this review at the onset – I am not a geophysicist. Hence, I cannot critically assess the dataset itself, the geophysical approach, or data processing, that has been used. I cannot completely assess whether the interpretation is appropriate given the data provided. As a result, this review is from an ‘external’ point of view with wider knowledge of the Superior and other Archean cratons. Hopefully other reviewers have the expertise to assess the geophysical dataset.

Decision

The paper is very interesting and provides some important insights. It has the potential to be an important contribution to the discussion on Archean geodynamics. However, currently it is too ‘one-sided’ and the interpretation needs to be stringer, and more robust, to prove their point(s). The pros and cons of alternative scenarios need to be discussed more fully, and more summary/schematic figures are needed to clearly convey their ideas to the reader. At the moment the figures are excellent but very technical.

As a result, I recommended the manuscript undergo major revisions. More detail is needed on numerous points, and the ‘identity’ of the paper needs to be clearer. However, I do want to clearly state that I think this work is very valuable and of high impact, and hence has a place in Nature Comms.

Summary of major points

- More figures to backup the arguments and models – at least one schematic figure clearly showing what they consider ‘microplates’
- A figure showing how they see the temporal evolution of the Abitibi going, in regard to their model/data, would be useful i.e., how does the late multi-phase compression impact the model?
- The maps and mofo models broadly match the isotopic maps of Mole et al (2021) – more links to this dataset could reinforce your arguments.
- Are microplates required to explain your data? It seems overly complicated to me and not really scale-realistic... could these simply be interconnected rifts that split and merge, just like in Iceland? (see below)
- When I envisage microplates – I see a complex network of subducting plates and triple junctions, like in Japan, or the Lau basin – is that really justified here? Some of the plates would need to be very small - <60 km in some cases? (the cartoon/schematic diagram would be very helpful here);
- Other models that come to mind are:
 - Iceland-like
 - Interconnected back-arc rifts
 - Asymmetric rifts/back-arc?
 - Basin and range style system?

[REDACTED]

From Karson et al (2017):

<https://agupubs.onlinelibrary.wiley.com/doi/full/10.1002/2017GC007045>

[REDACTED]

Thordarson and Larsen (2007)

[REDACTED]

Tectonic sketch of the Iceland plate boundary zone (PBZ).

From Karson et al (2017):

<https://agupubs.onlinelibrary.wiley.com/doi/full/10.1002/2017GC007045>

- You have 41 refs, but Nat comms allows up to 70, hence I have added a number of suggestions. The number of figures and refs indicates this ms may have been transferred from an early submission. This is totally fine, but at Nat comms is quite different, some re-writing and additions can now be made, and I think should be for this ms.
- Lean more on the geochemical and isotopic mapping that has been done – it supports your arguments/maps – you can use these to extrapolate to other cratons (i.e., Yilgarn)
- I don't think your identification of rifts or 'triple junctions' means that horizontal tectonics (plate tectonics?) was necessarily in operation – I don't see why these features couldn't occur in a stagnant lid or not plate environment (see details below).
- As I describe below, I am not sure this paper needs to get into the Archean geodynamics controversy – the data it has on rifts is of high impact enough. From what I can see, it could be used to make arguments for multiple settings, and does not support the existence of plate tectonics, perse.
- I think it might be good to discuss various settings, AFTER presenting the main finding of the rift setting for greenstones. And then, discuss the pros and cons of the different settings. What might be the best modern analogy for the setting you image? Present multiple comparisons if needed. You have up to 5000 works in Nat comms.

- Use the geology to support your points – the occurrence of komatiite is helpful. Petrologically they can only form from plumes, ambient mantle is too cool, even in the Archean (see Herzberg and Nisbet papers). Where are the komatiites? I think you'll find they match up with your rifts quite nicely (most).
- Remember, there is good evidence that these rifts were continental in nature – how's does that effect your model? (syn-volcanic zircon from these rifts do not suggest an oceanic origin, as well as rare, but constrained Mesoarchean zircon inheritance);
- A diagram that summarises your findings and clarifies the scaling – i.e., can you infer plate tectonic scale features based on the scale of features you have mapped? If so, which environments match up?

Further points

Line 10: why start the paper with this assumption? It is stated like a fact, but whether plate tectonics was indeed occurring in the late Archean is still debated. It is important to be mindful of the genuine debate around this – it should not be presented as fact.

Line 27: This model has been published by others for the Superior and other cratons and should be cited, for example:

Bedard et al <https://doi.org/10.1130/G35770.1>
 Mole et al <https://doi.org/10.1016/j.earscirev.2019.04.003>
 Strong et al <https://doi.org/10.1016/j.precamres.2023.107096>

Line 28: Really? Why? Why can't weak crust sustain rifting? I'd think weak hot crust would be more likely to rift, or collapse, as shown in Calvert et al:
<https://www.nature.com/articles/s41561-018-0138-0>

It will flow laterally, which could possibly aid rifting.

Line 32: Again, be careful stating debated points as facts – we simply don't know this yet. Many people think subduction started in the Hadean, others that plate tectonics was still not active at the end of the Archean. What you say is a general consensus, but one of the issues with Archean geoscience is these blanket 'factual' statements. I understand what you mean, but please be honest about the complexity and lack of certainty we face.

Line 36: Rifts do not necessarily = mobile lid. You can have local, or even large scale rifting or extension that does not mean 'plates' are moving as in a mobile lid. This is another assumption. The activity we see on Venus demonstrates this.

Line 37: I think this comment is unfair to other workers who have discussed rifting in the Abitibi. In the last few years there have been isotopic maps produced and published for the SE Superior that explicitly image and document large-scale rifting. Say these are 'unknown' is not really accurate. You could say 'have been previously imaged by geochemistry, but modern geophysics is yet to physically define these' – this is important, as your dataset supports the inferences and images made by the geochemical data, and you should lean on that to add evidence to your argument. It is great that these two completely independent datasets are showing such similar features.

Line 50: presuming these topographical features have survived 2.6 Ga of modification? What is the impact of younger magmatic and geodynamic events? Can you be sure these features are Archean? (this is where comparison to time-resolved isotope data might be helpful)

Line 62: This begs the question – why? Why is the mid-lower crust not deformed or affected by this? Is this normal in ‘orogenesis’ or major compressive events? Is this different to what we see in Phanerozoic orogens? This all feels relevant for a differential assessment of the Abitibi crust vs modern systems.

Line 83. This begs the question – has the geophysics simply imaged the surface features? If I understand correctly, your argument is that shallower moho will have higher gravity due to underlying dense mantle, whereas deeper moho will give the opposite. So, if the data correlate so well with surface geology, is it possible the intrusive (felsic, less dense), and volcanic areas (mostly basalt, more dense) are creating the appearance of belts of deep/shallow moho? I know this is hard to untangle, as you’d expect more volcanic/basalt rocks in the rift zones anyway, although not completely clear why intrusives would dominate the flanks, although these may be younger than the rift system, which supposedly ended around 2695 Ma.

Line 93: Sorry if this is an ignorant comment – but if you have 3D data, why do we not have any 3D images? What is better about the 3D data?

Line 94: I would argue here that you have added further resolution, and *confirmed* the architecture, already proposed by others. This is not the first rift architecture map of the Abitibi – your new work is very impressive, you don’t need to make these statements.

Line 116: this is a very quick mention of TTG petrogenesis! The question I’d ask is why this has happened more in the flanks? There are syn-volcanic plutons of various sizes all though the various zones of the Abitibi. Perhaps it is more about crustal level? If the rifts are dominated by volcanics, perhaps this is because they were localised into grabens, which filled with volcanic material and form a lower crustal level. The plutons are represented by the felsic intrusion dominated flanks, which are horsts. These areas also likely had volcanics, but due to rise of the horst and subsidence of the graben, the horsts expose a deeper crustal level than the grabens. It follows that there are likely felsic intrusions under the graben (and indeed there are many synvolcanic intrusions in these ‘rifts’), they just aren’t exposed yet. This is important because it’s an external support (from the geology) for the rift model. It’s important because many people think we cannot image the rift architecture like has been done here and isotopically.

The next stage would be to look for possible metamorphic grade variations across some of these zones to test this.

Line 119: yes but does it reflect the *thickness* of volcanic rocks?

Line 122: But these represent inverted rifts, right? They have been closed by the N-S compression and then translated to some degree by the strike-slip E-W deformation. Hence, can we presume these rifts would have been much wider, and what we are seeing is the inverted, closed versions? The same goes for the ‘depth’ of the moho – it is presumably a minimum?

Line 139: This paragraph is unclear – are you saying the rifts are not continental crust? If so, are you saying these are mafic crust? i.e., oceanic crust? So so, what is the basis for this? Can it not just be rifted or thinned continent? Not a big deal, but please clarify what crust you think is where, and why.

Line 152: this could be due to the lateral ‘flow’ of hot mafic crust (see the Karsens Iceland paper above), where the lower and mid-upper crust can become dissociated.

Line 158: I think you have missed a step here – you have imaged some rifts in Archean crust – now we are jumping to actual discrete plates and triple junctions. For me, the scale of these processes and features are not equivalent. If you disagree, you need to provide very good evidence that the rifts you have imaged can be equivalent to microplates. For a start, how would these mafic microplates exist in the Abitibi, which has continental crust back to at least 2750 Ma and possible longer.

Line 160: why are we only hearing about this oblique zone now? This should be documented in the Results above. In fact, a more detailed discussion of the features of the maps is warranted, especially in the scope of greater word count available.

Line 166: how continuous are these rifts? In many cases they seem to be interrupted. I am noticing a roughly similar scale with the Icelandic rift arms (~150 km).

Line 168: you split this up, but it would also be good to see it presented as one continuous system – i.e., a dismembered or fragmented rift, separated possibly by strike-slip systems (which could have localised deposits?)

Line 171: you are not the first to say this – please reference these points that are not original.

Line 175: This is the interpretation of Mole et al 2021;2022 – please reference original ideas.

Line 175: Note the occurrence of these kind of transform systems in Iceland (again Karsens paper above). In Iceland, they create a similar pattern of rifting and offset the main onshore rift with the MOR. Also the sites of major seismic activity, would could be linked to mineralisation: https://pubs.geoscienceworld.org/gsa/geology/article/32/9/813/103751/Fault-segment-rupture-aftershock-zone-fluid-flow?casa_token=u9E787KOo0MAAAAA:TGvkAsXs79QM1b8w3K86ILaFO84NqZhCqq6jhGCqXiykwfmVzSgjLun2go6EVklBy4X0bWo

Line 180: I would again reference the isotopic mapping which imaged this feature and supports your identification of it.

Line 180: A triple junction is a place where “three lithospheric plates meet”. Are you saying these rifts are discrete plates? If so what is the difference between this and extended and rifted continental crust, like in the basin and range? I think you have excellent evidence for a complex rift system in the Abitibi, but can you clearly demonstrate these rifts separates ‘plates’? One issue is that, isotopically, and geochronological, there is evidence of pre-existing crust in the Abitibi, so how does this factor into your model? Triple junctions are usually between oceanic or mafic plates...

Line 180: you mention horizontal tectonics in the abstract – if you are going to make that link, here would be best.

Line 183: The link to ore deposits is very interesting, but it does seem a bit out of the blue – I suggest adding more references to it throughout the abstract and intro so it doesn’t appear as a ‘tack on’.

Line 195: this is the first time you mention ‘oceanic rifts’ – so you think these rifts contain genuine oceanic crust? I think you need to be very clear early on what crust you think is where, and what kind of rifts these are. Remember continental rifting is also a thing...

Line 196: you have not discussed to any extent the nature of the crust previously, as arc-like or not. I wouldn't go near any controversy when you don't need to – just reference it as continental crust, or felsic crust. Don't be genetic if you don't need to, especially in the Archean...

Line 204: not 'would', use 'may have been'... do not operate in absolutes unless you are sure – and we are not...

Line 198-204: What are you saying here? Please be more direct. Are you saying you think the rifts formed by plumes, or by subduction? Make a case for one or the other, or both?

I think from what you have said here that rifting is not characteristic of subduction or plate tectonics. I don't think you need to get into the Archean geodynamics controversy with this paper and dataset, to be honest. I think it is totally ok to say that 'the rifts we have identified could be indicative of a number of settings...' and then discuss them and choose your preference if you have one.

The mistake you are making here is in assuming by the late Archean that subduction was operating, so the rifts must be related to subduction, hence this is evidence of horizontal tectonics, which is a very circular argument.

In my opinion, the identification of the rift architecture, showing that greenstone belts and possibly cratons are built based on a rift architecture, is big enough news for nature communications. I would argue it demonstrates that subduction is not required for these settings, with the architecture and geology all best explained by an Iceland scenario where a plume impinges on thick mafic crust, possibly with a slow spreading centre above. But a rift does not = plate tectonics or 'horizontal tectonics' in my view – subduction does. It also doesn't necessarily = a mobile lid. The 'triple' junctions also occur in Iceland, as shown above.

Line 207: I think the end of the paper is quite weak compared to the rest. Line 20 in the abstract is very strong in linking the data to plate tectonics and 'plate boundaries' but the end of the paper is not a decisive or clear on this. All in all the end section is quite vague. It is almost like the abstract decided to make some bold statements but the paper then (rightly, in my view) decided to tone it back.

Summary thoughts

I think the identification of the linear rifts, spacings, and NE-SW feature, is very cool. But I do not agree that the identification of rifts, which can occur in back-arcs (basin and range, Hellenic arc, Lau basin) and OIBs (Iceland) can be used to define a plate tectonic setting. I also think they can occur in non-mobile lid regimes. Intra-continental rifts would be an example, often formed during LIP emplacement. However, I do think this is an important finding regarding the environment that cratons form and greenstone belts form. Archean rifts have been inferred by numerous datasets and mapping data, as well as mapped using isotopic data for the Yilgarn and Superior but showing them directly using geophysics has been massively hampered by the late compressional events.

This dataset is the first to *directly* image Archean rifts, proving the hypotheses suggesting by the geochemical mapping and lithological constraints. Remember, TTG-like rocks and numerous felsic volcanic centres are present in places like Iceland also. I think the authors should focus on that point more. To be fair, the main paper does, it just ends without being very sure of itself, and the abstract sets it up for something it doesn't deliver.

I think if the abstract is re-written, and the paper edited to be less ambiguous on its general findings, it will make a valuable contribution to the science.

Figures

Figure 1

Figure 2: include the Abitibi outline on this – I would also recommend adding some key locations, places or mines, for every figure so there are known ref points for readers. Most people cannot place these data without some kind of reference frame.

Figure 3: The red-blue colour profiles used are not generally good for colour-blind people and may not present data clearly (see <https://www.nature.com/articles/s41467-020-19160-7> Crameri et al 2020). I am not sure the colour palette is helping us see the variation in 3b. You might want to consider changing these and being consistent across all figures.

Methods

I have not critically assessed the geophysical methods due to my lack of expertise – I leave this to other more qualified reviewers. However, this assessment is critical to ensure the dataset is robust. We encourage the authors to ensure all raw data are available for others to repeat and test their model.

Review of the revised “Archean rifts and triple-junctions revealed from gravity modelling of the Superior Craton” by Galley et al

Introduction

The revision of this paper by Galley et al represents an improvement on the previous version. The majority of the revisions are excellent – however a few further comments are provided below. This paper deserves to be published in Nat Comms, but it needs to be more exact in its points in key places. Parts of the paper and its major points on tectonics are still far too vague.

This is a good paper, but it could be a *great* paper with a bit more work and care and attention. There are numerous typos and errors, and the prose itself could be clearer in places. However, the science is sound and I think this will be a really nice contribution after these revisions. I envisage that after this relatively straight-forward revisions the paper will be suitable for acceptance.

Response to previous comments

I do not particularly like the way the authors have summarised the reviewer’s (my) points and then made their comment or rebuttal. Please paste the *actual* comment, verbatim, into your report, and then address it. This avoids any ambiguity; the editor and I can see the original comment next to the response.

In general, the revisions were thoughtfully performed and comments addressed or rebutted well. However, I have further questions regarding the responses below:

- Thank you for adding figure 5. This is excellent and I think is a very powerful link between the geology, geochemistry, and geophysics. This is something that is, unfortunately, quite rare. I suspect this figure will be used a lot.
- I still find the microplate argument ambiguous. Were these pieces of crust actually totally separate plates? Or different terranes? I am not sure... but I think calling them microplates will at least generate some interesting discussion. Your definition is progress, but the rifts surely have to be ‘mantle tapping’? i.e., like at plate margins? I wonder if the situation in East Africa is a good analogy. Some discussion on the size and geometry of microplates needs to be in the main text. Apologies if I missed it.
- You say “the abstract is re-written” but the track-changes show only four changes that are extremely cosmetic. The two statements you make in the abstract are not factually correct (lines 9-10 and lines 20-21) and these need to be revised or toned down. It will not affect your ms as far as I can see but be more accurate to current Archean geoscience. The reality is, we cannot be sure of many things in Archean geology and geodynamics, and hence we need to be open about that as we try to push things forward.

Read-through of revised ms:

- I think I made this point in the first review – the first line of the abstract is not appropriate (line 9-10). It makes a statement as fact, that we do not know. We do not know that plate tectonics started at 2.7 Ga – this is not a fact and needs to be removed or edited. I’m sorry, but I insist on this. This is because there are a whole range of opinions on when PT started, from the Hadean, Mesoarchean, Neoproterozoic, to Neoproterozoic. Even if you mean “non-subduction, mobile-lid plate tectonics”, this statement will not necessarily be read this way – if you do mean this, you should at least clarify it. Furthermore, this would also be debated – some think a stagnant lid is appropriate for the Neoproterozoic. I do not see why microplates and rifts could not exist in a stagnant lid, with localised extension taken up by ‘crumpling’ at the margins, similar to Phanerozoic ‘intraplate orogens’ such as the Alice Springs

Orogen. Again, this is what is theorised for Venus. A connection of rifts does not necessarily mean moving plates – if you want to infer this, you probably need further evidence. Referencing some paleomagnetism from Archean rocks would be a start, such as these papers:

<https://pmc.ncbi.nlm.nih.gov/articles/PMC7176424/>

https://www.sciencedirect.com/science/article/pii/S0012821X0700516X?casa_token=DtodWmKuBKkAAAAA:m3VPwWcldcMmDoZTCtttjHUPMWgYhWKBOT2fOHDYOji_2PxuoNDRNMycU4YqaBYfDIXx8lyDA

They show evidence of plate movement in the Archean, although as you can imagine, this is contested. I personally like the idea of episodic subduction, and this matches the data in Mole et al 2021 for the SE Superior. However, these rifts appear to be 'pre-subduction', if subduction started at 2.695 Ga in the Abitibi, so in that context they would potentially need another driver.

On top of this stuff, some of the hottest magmas on Earth are known from the Neoproterozoic, which suggests this may have been a thermal peak, rather than a cooler inflexion (see Herzberg et al 2000). You can say this, but not as a *statement*. You need to add 'may', or 'possibly' etc. If you feel this is a result of your study, that's fine, but don't open with it. Right now, as I said in the previous review, this is not correct, and a poor way to start the paper.

You might rather say that *"The nature of Archean global tectonics, and the associated geodynamic regime, is much debated in modern geoscience, despite decades of research. In this study, we use a new geophysical dataset to show that, by the Neoproterozoic, convective forces etc..."*

That way you can state lines 11-14 more as your result and findings, which they are.

- Line 18: timing is important for the Abitibi – is this model strictly for the syn-volcanic period i.e., 2750-2695 Ma? If so, a few lines on how the compressive phase 2695-2670 Ma influenced it might be useful to complete the circle. You say in the abstract your microplates are evidence for horizontal tectonics (i.e., compression) but there is no evidence for this structurally until ~2690 Ma – how does this work?
- Lines 19-20: You have found rifts – or plate boundaries in a mobile-lid tectonic setting – this not direct support for horizontal tectonics. Archean tectonics is littered with these kinds of statements. Report on what you *actually* have – rifts are evidence for extension. Horizontal tectonics will be interpreted as subduction – if you don't mean this, then state 'mobile-lid' or whatever you do mean.
- Lines 25-26: you have not referenced this. Please be more exact when describing what you mean by plate tectonics – just mobile lid or also subduction? – if the latter, say 'modern-style plate tectonics'. Rifting does not = modern-style plate tectonics, so don't infer that. If you mean mobile-lid and horizontal movement – then be specific. Don't leave ambiguity. This a massively important point – be very clear on what you mean.
- Line 35-36: I do not understand why you keep making statements. There are very few things in Archean geoscience that can be said with certainty, as a statement. This 'may' have been a pivotal time – we don't know for certain – new data suggests the Neoproterozoic was pivotal. Don't state facts that we don't have for the Archean. Half of these statements would be corrected if you just inserted 'may' or 'likely'. It's a

problem in Archean geoscience that two papers next to each other can make two contrasting absolutist statements that contradict each other. It's made the science messy, don't be one of them.

- Ah ha! Line 43 we get to the point! So, when you say mobile-lid, you DO mean mobile-lid +/- subduction. You should remove references to 'horizontal tectonics', and just say mobile-lid. Horizontal tectonics is disliked as a term by many (JF Moyen for one) and it is mostly associated with modern-style plate tectonics. You can remove all ambiguity by simply stating 'mobile-lid' tectonics. Many see horizontal vs vertical as problematic language, when it is likely to be a dynamic continuum.
- Of course, relatively small-scale rifts of this nature do not necessarily mean there was a global mobile-lid either... but at least local mobility. On Venus rifts are taken up by localised folded mountains, a system known as "*tesserae*" – these are regional dynamic forces in the mantle, and not associated with a true mobile-lid. In fact, you should probably mention such features as part of your interpretation – they are a valid interpretation for what we see in the Archean. I think it would be really valuable to clearly present all these options, even as a list if it's easier, and then choose your preferred option (with reasoning). You have done this in the 'origin of rifting' section, but you could do more. I suspect you have more words available – don't be afraid to add more detail.
- Line 50: parts of the Superior are younger than 2690 Ma.
- Line 58 – spelling error 'ruft'
- Line 64 – change to 'mobile-lid'
- Line 85 – Mole et al 2022 should be here, Mole et al 2021 should be at the end of this line. Mole et al 2021 is the original source of the isotopic and WR dataset. Use Mole et al 2022 for any mineral systems inferences. Some mistakes across the ms with these references.
- Line 119-124: It is good you have made this point, but it feels like a 'we hope this isn't the case' – can you be more specific? It seems a key point – can you prove that we aren't just seeing the surface effects? Apologies if this is already in the Methods. Maybe a reference here would help.
- Line 131: if the resolution of your model is 10 km, and the range in moho depth is 32-45 km (line 149) – this range is very close to the uncertainty... does this affect how resolvable the features are?
- Line 153: partial melting of the crust
- Line 161: good point
- Line 175: Matheson
- Line 180-181: length is used twice here, re-write
- Line 183: time? (spelling)
- Line 183: some actual examples here would be useful, rather than just citations.

- Line 185: hmm is this true? A stronger argument might be that the Archean crust was hotter, and therefore behaved differently during compression – see the paper below: <https://www.nature.com/articles/s41561-018-0138-0>
I suspect that the rift zones were more likely to survive in a crust that collapsed laterally, rather than thickened vertically. This might provide some better evidence for your point.

I'm not sure if you mention it, but it seems intuitive that the width of your mapped rifts are minimums, and that they have been closed at least partially by the later compression.

- Figure 5: As I said – this is a really nice figure. Combining the separate systems endorses your model and validates those existing models into a holistic system. However, for some reason you have cited Mole et al's Yilgarn paper for these data. The correct reference should be the Mole et al 2021 paper (Precam Res). Mole et al references seem to be mixed up a bit in their usage: The isotopic/geochemical dataset itself is from Mole et al 2021. When using it to refer to mineral systems, use Mole et al 2022. If you want to compare your dataset to similar systems in other cratons, or cite an example of other Archean rifts, use Mole et al 2019 (i.e., Kalgoorlie-Kurnalpi Rift; KKR).
- Line 214: be careful when using 'most reasonable' – I know what you mean, but unfortunately what is reasonable is subjective... Better to say, something like, 'in conjunction, the geophysical and geochemical evidence best supports a rift model....'
- Line 230: apologies, you have already seen the Calvert paper. I still think you might look to use it in the point I mention above. Apologies if I already said this stuff last time!
- Line 279: you need a reference here
- Line 295: "Ey" should be Eu
- Lines 292 to 295: you have these Eu/Eu*/Y relationships backwards. Values >1 indicate a deeper source/thicker crust (possibly – some contention on this) and <1 shallower crust. This is because, for deep crust we expect high Eu values (no plagioclase), and low Y values (garnet present). The high Eu = higher Eu/Eu*, and lower Y values result in a higher final number. Hence deep crust = >1. Vice versa for shallow crust with plagioclase present, which takes out the Eu, no garnet = higher Y. See below figure from Mole et al 2022. You seem to have this correct in Fig 5D, so just an error in the text I think. Note the U/Yb map in Mole et al 2022 correlates quite nicely with your N and S rifts.

[REDACTED]

- Line 291-298: This section is good but feels rushed. You talk about La/Sm, but in the Figure 5D, you have Sr/Y labelled (which is generally a depth indicator). Please make the figure and text consistent.
- The low La/Sm could also indicate less crustal contamination i.e., from passing through thinner or no silicic crust. Contamination indicator (Th/Yb) correlates with decreasing eNd in Mole et al 2021, possibly suggesting the trend is due to contamination with an older, existing crust.
- Line 307 – again clarify what you mean when referencing horizontal tectonics. This last section is where you need to be very clear what you are inferring; is it mobile-lid tectonics? Papers by Gerya et al have shown how plumes can form mobile lids, which then induce short subduction events – this might fit the timeline of events in the Abitibi:

[REDACTED]

- Origin of rifting section: I know what you are trying to say here but I think it could be more clearly written. I think removing vertical vs horizontal tectonics will help it be more specific – talk about the processes you mean directly i.e., sagduction/drip tectonics and the other options discussed in this response. If you are talking around mobile microplates, then say that. You have done a good job and trying to extricate the data from going too far, which is great, but I think it needs a bit more polish. For Nat comms, you need to provide more, just saying it means horizontal tectonics is too vague – show us how well you understand the complexities of Archean geoscience and go further with your model/process.
- Remember, just because you have evidence of regional mobile microplates does not mean it was a completely mobile-lid – as I said before, Venus has evidence for this and is generally seen as a stagnant lid. I think ‘locally mobile-lid’ covers pretty much all bases. In the plume model above (Gerya et al), the rifting is actually driven by a vertical process, with lateral spread of the ‘plates’ – so I don’t think committing to horizontal tectonics is necessary or even correct at this point. I think you could include this, or at least present this further alternative.
- If you *must* use “horizontal tectonics” at least use it *after* the explanation you provide. That would mean removing it from the abstract to avoid confusion.
- Line 320-321 – you use ‘although’ and then ‘however’ in next sentence – these are synonyms, reword this.
- Line 321: why? What about the model suggests this?
- Line 323: don’t need ‘widespread’ here, it’s ambiguous and I’m not sure you have evidence for it.
- Line 323: This is not the best way to end the paper. If you have words left, a paragraph bringing the reader back to the main point, and its implications, would leave a better impression. Perhaps use the last sentence to summarise the discussion of different models/ideas from the section above? Or leave the reader with what further implications may be? i.e., “ *our results show a strong empirical relationship between mantle plume activity, microplate formation, and major mineral systems for the Abitibi, which imply at least a locally mobile-lid tectonic environment existed in the Neoarchean. These results have implications for the formation and evolution of all cratons, as well as their host mineral systems.*” Something like that?
- I think it would also be worth making the point that deep sea hydrothermal systems have been associated with these rifts, as exemplified by the VMS systems. The origin of early life may be connected to these type of settings, and hence these new maps of Archean rift systems map out potential locations for early life i.e., <https://www.sciencedirect.com/science/article/pii/S0301926814001454>
This might be a nice way to end the paper, to show it has profound implications for the wider Earth system and environment.
- There is also a new paper in Geology (screen shot of a figure below): <https://pubs.geoscienceworld.org/gsa/geology/article/53/3/269/650953/Inception-of-ridge-ridge-ridge-triple-junctions>
It might be good to reference and/or include this evidence for rifts in an arc environment, in the interest of balance. It doesn’t mean you need to commit either way, but it would mean you present possibilities for multiple settings. It also supports your ‘mobile-lid’ / ‘laterally-dynamic’ tectonic process. Perhaps with the komatiites occurring at the same timing of rifting, and komatiites connected to plumes (ambient mantle was too cool; Nisbet et al.), the plume model is more likely; but you should decide what you prefer.

[REDACTED]

Major changes required:

- Remove all occurrences of 'horizontal tectonics' – replace with what you mean, which is 'local mobile-lid tectonics'. I know what you mean, but horizontal tectonics as a term is closely associated with subduction and modern-style plate tectonics, so better to use something else.
- There are other possibilities you need to explain and explore, to make the origin of rifting section more thorough and less vague.
- As stated in the first review, do not make statements that are not currently valid in Archean geoscience. The reality is we don't know enough on many aspects to make statements – for most points there is a counterpoint. This is easily fixed by qualifying statements by adding 'may' or 'likely' to take the absolutism out of the sentence.
- Tidy up the quality of the writing – it feels a bit rushed. There are numerous spelling mistakes, and some sentences could be clearer. Be very careful when presenting geochemical data – there are some major blunders in there.

Decision

Despite the important changes I'd like to see, they should be fairly straightforward. If the authors are willing, these will be easy to do. Hence the decision on this revision is ***moderate-minor corrections***. These revisions can be assessed and approved by the editor, and will not require re-review, if completed fully. Once revised, I see no reason why this will not make a valuable contribution to Archean geoscience and of interest to the readers of Nature Communications. On a personal note, I like the paper and think it will spark some good discussions – great to see this quality of work from an ECR.

Review of “Archean rifts and triple-junctions revealed from gravity modelling of the Superior Craton” by Galley et al.

This paper uses gravity anomaly data to infer the depth to Moho in an Neoarchaean greenstone belt in Canada. The main result of the paper is to show that there are “corridors” of shallow Moho within the belt and these reflect thin crust associated with rifts and possible evidence of plate boundaries, including triple junctions. If this is correct then these results suggest that plate tectonics was operative on around 2.70 – 2.75 Ga, which is close to the ~2.80 Ga limit of most previous studies. Before that Earth has been considered by most authors as being too hot for cold rigid plates to exist, the motion of which could be described by rotation about Euler poles.

I have a number of concerns/queries about this paper.

1. Line 49. There is an assumption in this paper that studies of today’s gravity field can be used to infer crustal structure in billion-year-old cratons. This assumption should be stated and justified. For example, there is evidence from early studies that the boundaries between cratons in Canada are associated with distinct free-air gravity “edge effect” anomalies (e.g., Gibb & Thomas, 1976, Nature). This observation supports the notion that the crust and upper mantle are essentially elastic and can support large stresses on long geological timescales.
2. Lines 68/69. The statement that “Inversion modelling creates a petrophysical model” is misleading. Inversion in gravity interpretation is a procedure that creates a body with a certain specified density contrast with its surroundings *directly* from an observed gravity anomaly.
3. Line 71/72. Why is it relevant that the *vidi* inversion code was run on Canada’s *narval* cluster? Is the code limited to certain platforms? Is it publicly available?
4. Line 73. Give the reasons why the “3.07 g/cm³ iso-density surface” was selected for the Moho. The value, which is lower than that assumed at a mid-ocean ridge, seems low in comparison to densities assumed in studies of the cratonic mantle elsewhere (e.g., Africa, Brazil).
5. Line 78. Explain what the “signal” is that was output from the inversion code. The measured Bouguer anomaly is referred to the geoid and assumes in its reduction a certain density for the rock between the physical surface and mean sea level. Was the “signal” similarly referred to sea level?
6. Line 100/101. The Moho derived from the inversion model is compared to CRUST1.0, a teleseismic model and five Lithoprobe seismic lines. However, there are issues associated with each of these sources of seismic data. CRUST1.0 is a compilation of existing active source seismic experiments from around the globe which have been gridded at 1 degree (i.e. ~111 x 111 km) intervals. There is no evidence provided in the paper that there is any active source data available within the region of the study area. The teleseismic data is limited to Receiver Function data which is based on an assumption of *P* wave crustal velocity between the physical surface and the Moho. There is no evidence provided in the paper of what velocity was used or discussion on whether or not this velocity is an appropriate one to use in the study area. Finally, the Lithoprobe data are seismic reflection profiles and so are in two-way travel time, not depth. There is no evidence provided in the paper of what velocities were used to convert the reflection data to depth.

7. Figures 3 and 4. The relationship between the observed Bouguer anomalies and the inferred Moho undulations is very difficult to see in the paper. At the very least profiles of the observed Bouguer anomaly should be shown in both Figures 3 and 4 so that the reader can see the relationship.
8. Line 105. Was it not assumed in the model that the sub-surface density structure was layered?
9. Acknowledgments. Makes no mention of data availability or access to the inversion software used in the paper. Web links should be provided that link to the Bouguer anomaly and topography data and to the *vidi* inversion code.

In summary, this is a paper on an interesting topic related to the structure of continental cratons. However, I have many concerns about the validity of the results presented in this paper. Perhaps the most concerning was the lack of any verification of the inversion results using forward modelling techniques. For example, the gravity effect of each iso-density layer should have been calculated using an independent methodology (say by using Parker's Fast Fourier transform) and compared to the observed Bouguer gravity anomaly on a grid. This is because the modelled Moho undulations are small and it is not at all clear that the Moho needs to undulate at all. The authors therefore need to demonstrate that a model in which the Moho is flat throughout the study area and the layered density structure is limited to the crust cannot explain the observed Bouguer anomaly as well as a model in which the Moho undulates.

These and the other recommendations made in this review will, in my opinion, take some time to address in full and so I recommend rejection and re-submission of the paper in its present form.

I have looked over the revised paper of Galley et al. Unfortunately only a few of my comments have been addressed.

1. The assumption of 3070 kg/m^3 for a mantle density has still not been justified. The density, which is more typical of lower crustal is too low for the mantle – it is even lower than the density (3180 kg/m^3) normally assumed for the mantle at an actively spreading mid-ocean ridge!
2. The inverted Moho is still being compared to travel-time seismic reflection data. There should be an explanation of how these data have been converted to depth and what the uncertainties on these depths might be. Readers should not have to search through the literature for this information.
3. The authors have now provided a Bouguer anomaly profile, but the vertical scale range of 1000-1060 mGal in Figure 4D makes no sense. I could not look up what the Bouguer anomalies are in the study area because none of the maps in the paper have latitude and longitude (they should do!) but Government maps show Bouguer anomalies in the range -20 to -100 mGal southeast of Hudson Bay.
4. The output of the inversion modeling is a density structure that has still not been verified by forward modeling. It should be a relatively straight-forward process to take the density contours such as those shown in Figures 3B and 4D and calculate the gravity effect of each contour using a line-integral method and the density contrast between layer above the contour and the layer below.
5. The source code of the computer program used in the inversion is still not being made publicly available. Readers will therefore be unable to understand how the code works or, more significantly, be able to reproduce the results in the paper.

In summary, I am not convinced that the undulations in the Moho proposed in the paper are required to fit the observed gravity data. The authors need to demonstrate more clearly that the sources of the Bouguer gravity anomalies are not within the crust and that Moho is not flat.

Review of “Archean rifts and triple-junctions revealed by gravity modelling of the southern Superior Craton”.

I have looked over the revised paper of Galley et al. Unfortunately only a few of my earlier comments have been addressed.

1. **The 3070 kg/m³ density contour.** I checked the Galley et al. (2024), Welford (2024) and Welford & Hall (2007) papers but could not find any reference in them to the 3070 kg/m³ contour as Moho. The Parker methodology is the most commonly used way to compute the effect of Moho because seismic refraction data indicate that the transition from crust and mantle is generally associated with abrupt change in density, not a gradual one. At the very least, a comparison should be made between the model predictions and one assuming the crust and mantle had homogeneous densities of, say 2850 and 3300 kg/m³ respectively.
2. **Comparison with seismic data.** The absence of controlled source seismic refraction data remains an important missing constraint on the gravity modeling. I am not convinced that seismic reflection profile data can provide any useful constraints on Moho depth in the absence of detailed velocity information. Teleseismic data can provide estimates of Moho depth, albeit for an assumed average velocity of the crust, but requires closely spaced seismic recording stations and techniques such as *H-k* stacking, which I do not know were employed in this case. Finally, CRUST 1.0 is a 111 × 111 km grid of existing, mainly seismic refraction, data which be of limited use in resolving the subtle, short wavelength undulating variations in Moho depth inferred in this paper.
3. **Observed and ‘processed’ Bouguer anomalies.** I am afraid that simply adding the word ‘processed’ to ‘Bouguer anomalies’ that are of the order of 1000 mGal makes no sense to this reviewer. The observed Bouguer gravity anomalies in the study region are approximately -20 to -100 mGal (according to Canadian government maps) and all figures that purport to show the observed Bouguer anomaly should reflect this.
4. **Forward modeling.** The modeling in Figure S4 is interesting, but it is not a verification of the density structure deduced from inversion by forward modeling. Rather, it shows the modeled contributions to the Bouguer anomaly of sources in the crust compared that of the mantle. Interestingly the figure, together with Figure 4D, shows how the inversion places so much of the source of the short wavelength gravity anomalies in the upper crust. The lower crust appears to make little contribution. Such a density structure for the crust in a cratonic setting is surprising and should be verified, for example, by spectral modeling. I was more confused, however, about what Figure S4 reveals about the long wavelength contribution. Is it mainly from the Moho (ie the crust/mantle boundary), the upper mantle or both? If the latter, then would we not expect a ‘trade-off’ between the two sources such that the Moho could be flat and provide no contribution while sources in the mantle provide the main contribution?
5. **Software availability.** I repeat my earlier statement that the source code for the inversion method should be made freely available to *all* researchers so that the community has the opportunity to verify the methodology and repeat the calculations made in this paper.

In summary, my concern about the gravity modeling remains. Therefore, I question the main result of the paper: namely that the regions of shallow Moho reflect narrow linear belts of thin rifted crust that are bounded by the broader regions of thicker crust of one or more Archean ‘microplates’.